# Dynamic Distribution of Gut Microbiota in Pigs at Different Growth Stages: Composition and Contribution

Yuheng Luo,[a] Wen Ren,[a,b] Hauke Smidt,[c] André-Denis G. Wright,[d] Bing Yu,[a] Ghislain Schyns,[e] Ursula M. McCormack,[f] Aaron J. Cowieson,[e] Jie Yu,[a] Jun He,[a] Hui Yan,[a] Jinlong Wu,[b] Roderick I. Mackie,[g] Daiwen Chen[a]

[a]Key Laboratory for Animal Disease-Resistance Nutrition of Ministry of Education of China, Key Laboratory for Animal Disease-Resistance Nutrition and Feed of Ministry of Agriculture of China, Key Laboratory of Animal Disease-Resistant Nutrition of Sichuan Province, and Animal Nutrition Institute, Sichuan Agricultural University, Chengdu, People's Republic of China

[b]DSM (China) Animal Nutrition Research Center Co., Ltd., Bazhou, People's Republic of China

[c]Laboratory of Microbiology, Wageningen University, Wageningen, Netherlands

[d]Department of Animal Sciences, College of Agricultural, Human, and Natural Resource Sciences, Washington State University, Pullman, Washington, USA

[e]DSM Nutritional Products Ltd., Kaiseraugst, Switzerland

[f]DSM Nutritional Products France, Centre De Recherche En Nutrition Animale, Saint Louis, France

[g]Department of Animal Sciences, and Carle R. Woese Institute for Genomic Biology, University of Illinois, Urbana, Illinois, USA

Yuheng Luo and Wen Ren contributed equally to this article. Author order was determined on the basis of contribution.

**ABSTRACT** Fully understanding the dynamic distribution of the gut microbiota in pigs is essential, as gut microorganisms play a fundamental role in physiological processes, immunity, and the metabolism of nutrients by the host. Here, we first summarize the characteristics and the dynamic shifts in the gut microbial community of pigs at different ages based on the results of 63 peer-review publications. Then a meta-analysis based on the sequences from 16 studies with accession numbers in the GenBank database is conducted to verify the characteristics of the gut microbiota in healthy pigs. A dynamic shift is confirmed in the gut microbiota of pigs at different ages and growth phases. In general, *Bacteroides*, *Escherichia*, *Clostridium*, *Lactobacillus*, *Fusobacterium*, and *Prevotella* are dominant in piglets before weaning, then *Prevotella* and *Aneriacter* shift to be the predominant genera with *Fusobacterium*, *Lactobacillus*, and *Miscellaneous* as comparative minors in postweaned pigs. A number of 19 bacterial genera, including *Bacteroides*, *Prevotella*, and *Lactobacillus* can be found in more than 90% of pigs and three enterotypes can be identified in all pigs at different ages, suggesting there is a "core" microbiota in the gut of healthy pigs, which can be a potential target for nutrition or health regulation. The "core" members benefit the growth and gut health of the host. These findings help to define an "optimal" gut microbial profile for assessing, or improving, the performance and health status of pigs at different growth stages.

**IMPORTANCE** The ban on feed antibiotics by more and more countries, and the expected ban on ZnO in feed supplementation from 2022 in the EU, urge researchers and pig producers to search for new alternatives. One possible alternative is to use the so-called "next-generation probiotics (NGPs)" derived from gastrointestinal tract. In this paper, we reveal that a total of 19 "core" bacterial genera including *Bacteroides*, *Prevotella*, and *Lactobacillus* etc., can be found in more than 90% of healthy pigs across different ages. These identified genera may probably be the potential candidates of NGPs or the potential target of microflora regulation. Adding substrates preferred by these target microbes will help to increase the abundance of specific symbiotic species and benefit the gut health of pigs. Further research targeting these "core" microbes and the dynamic distribution of microbiota, as well as the related function is of great importance in swine production.

**KEYWORDS** pig, gut microbiota, dynamic distribution, host phenotype, regulation

Address correspondence to Daiwen Chen, dwchen@sicau.edu.cn.

The authors declare no conflict of interest.

Similar to humans, pigs have a complex and diverse community of microorganisms in their gastrointestinal tract (GIT), that plays a fundamental role in immunity, physiological processes, and the metabolism of nutrients. The diversity, composition, and function of gut microbial community are influenced by various factors including diet, age, stress, and the environment (1). This can directly or indirectly affect the metabolism, immune response, and intestinal homeostasis of the host (2, 3), constituting a so-called "cross talk" between the gut microbiota and the host (4). Although the gut microbiota has been regarded as an important metabolic "organ," studies on the microorganisms in the GIT of pigs are scarcer than in humans. A recent study presents a collection of cultured bacteria from pig GIT and reveals very different taxonomic groups with special metabolic functions (5), suggesting that our understanding of gut microbiota in pigs is less comprehensive. The dynamic distribution and contribution of a healthy microbial ecosystem in pig GIT have yet to be qualitatively or quantitatively defined as a tool to help maximize animal health and growth performance (6). On the other hand, different feed additives, such as antibiotics (7–16), zinc oxide (ZnO) (17, 18), probiotics (15, 16), and prebiotics (9, 17–21), have been reported to induce shifts in the microbial community associated with growth performance, thereby providing new insights into helping identify functionally important microbes as prospective biomarkers that are beneficial for growth performance of pigs.

Here we first summarized the characteristics and the dynamic shifts of the bacterial community in the GIT of pigs at different growth phases based on the data reported in 63 published papers. A total of 57,875,211 16S rRNA gene sequences from untreated, healthy pigs from 16 different studies (from the 63 published papers) were downloaded from NCBI GenBank for a meta-analysis to verify characteristic microbial populations. Based upon this metadata, the contribution of the gut microbiota to growth phenotypes of the host were evaluated, in order to discuss functional microbes, and/or potential biomarkers in swine.

## RESULTS AND DISCUSSION

**The dynamic shifts of the GIT microbiota of pigs at different ages.** Age is an important factor driving the maturation of the gut microbiota (22, 23). The microbial diversity in the swine GIT shifts over time (10), and the dynamic distribution of the gut microbiota in pigs can be thought to present along a longitudinal age axis (Fig. 1A; Fig. S1). Colonization of the gut microbiota starts during birth, as soon as the newborn comes into contact with microbes from the mother and the surrounding environment, and is shaped by the consumption of the sow's colostrum and milk, resulting in a milk-oriented microbiome (12, 13). Nursery pigs (0 to 21 days) harbor the lowest microbial diversity compared to older pigs, and this is an important reason why piglets are more susceptible to infection and less efficient in the utilization of nutrients before weaning compared to post weaning (24, 25). The weaning period offers a special window for modifying the gut microbiota. Weaning occurs at 21 to 28 days of age in most commercial swine operations, and piglets are switched to a diet containing less digestible ingredients (such as cereals) instead of milk, thereby creating a sudden, complex, and highly stressful event. The resulting nutritional, physiological, and psychological stressors often lead to dramatic changes in the piglets' intestinal morphology, physiological function, and the microbial community, increasing the risk of severe diarrhea, or even death (14–16). Thus, these remarkable changes in the gut microbiota of pigs across different ages greatly affects the general nutritional or health assessment of pigs. In this case, it is meaningful to take the dynamic development of the gut microbiota into account at each growth stage.

**Development of the gut microbiota in piglets from birth to weaning (0 to 20 days).** Infancy is a critical period for microbial colonization, during which microbiota structure is unstable and susceptible to surrounding environmental conditions. Increasing evidence has demonstrated that the early colonization of "appropriate" microorganisms may determine the gut microbial composition and immunological maturation (20, 26, 27). The GIT of piglets is said to lack microbial colonization prior to birth. However, immediately

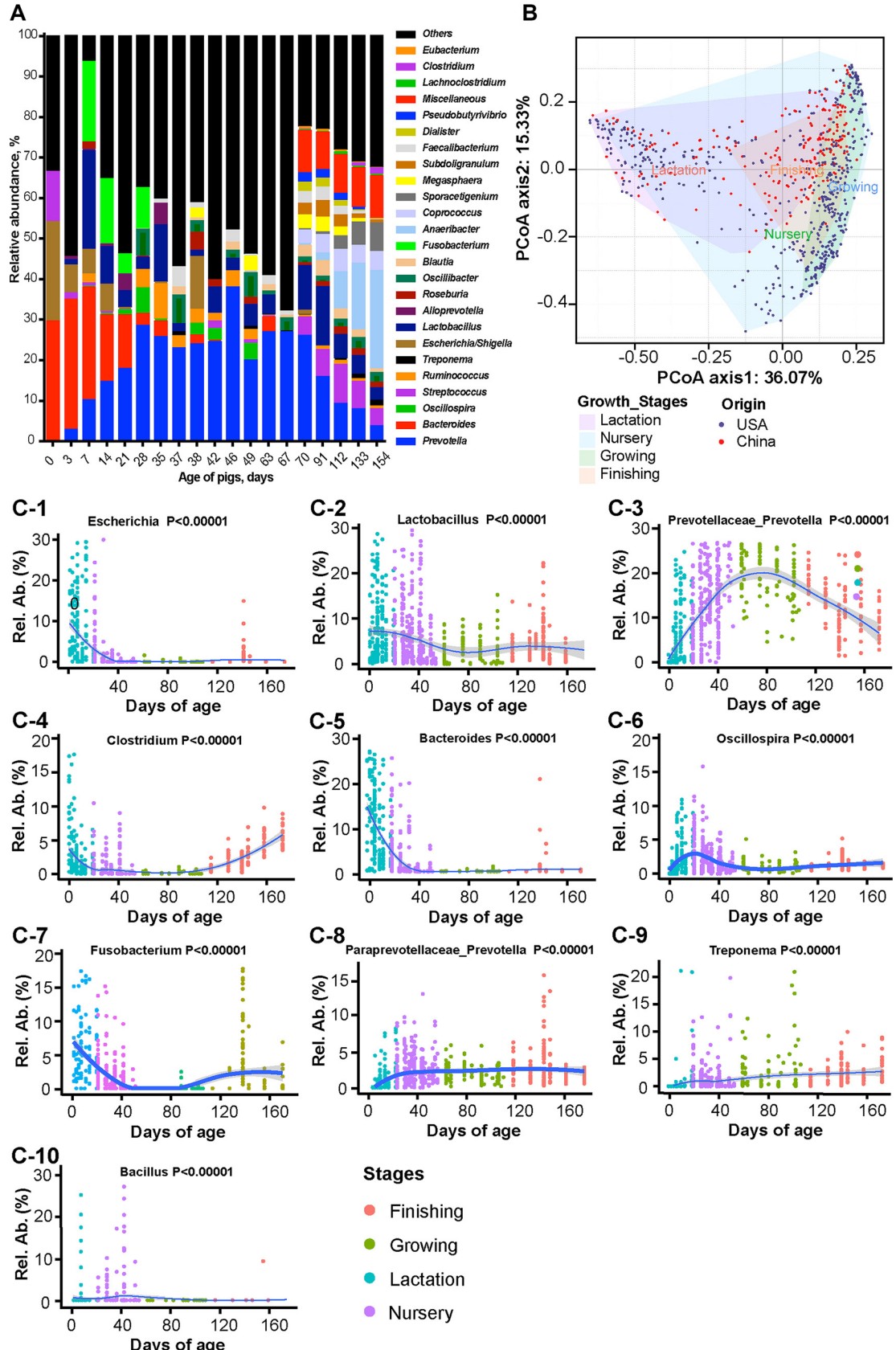

**FIG 1** (A) The dynamic distribution of bacterial genera in the gut of pigs at different ages. The relative abundance of the top 25 genera are shown. Data are summarized from 63 publish papers. (B) Longitudinal changes in the microbial community at

after birth, bacterial populations are immediately introduced to the GIT. This process is mainly influenced by their mother (28), the colostrum (29), and from the skin and feces, which is why the fecal microbial community of suckling piglets is similar to that of their mother (30). Recent investigation suggests that the early establishment of a stable gut microbiome plays a fundamental role in the development of the physiological function of the GIT, and the maturation of the innate immune system after weaning, which directly affects the growth performance of growing and finishing pigs (7, 31–34). The environment, especially the housing environment, is likely to be another source for the establishment of the gut microbiota in piglets, because co-housed piglets harbor very similar microbial communities (30). These studies suggest that microbes from the maternal and surrounding environments may play an important role in the microbial succession of newborn piglets.

The development of the gut microbiota in the piglet is gradual and sequential (33, 35). Sequencing data clearly showed the effect of age on the development of the gut microbiota from day 1 to day 21 (Fig. 1A; Table S1). In the first week after birth, *Bacteroides*, *Escherichia*, and *Clostridium* are the three most abundant genera. A marked shift from *Bacteroides* to *Prevotella* has been observed as piglets age/grow (33). In fact (Table S1), just 2 weeks after birth, *Prevotella* becomes the third dominant genus in the gut of piglets, and increases to become the most abundant genus at weaning (21 days). In the modern pig industry, piglets start to receive commercial feed 3 to 5 days after birth. The shift from sow milk to solid feed helps shape the GIT microbiota of piglets. Microbes in the gut of neonates are adapted to use a wide range of both milk oligosaccharides and host-derived glycans (e.g., sulfomucin) as their unique carbon source (36). Due to the ability to utilize multiple saccharides, *Bacteroides* can be found as a predominant genus in the gut of pre-weaning piglets, helping the host to efficiently utilize most of the milk-derived carbohydrates (27, 33, 37–39). *Prevotella* is associated with an increased long-term carbohydrate intake (40), and is capable of metabolizing complex dietary polysaccharides (41). With the intake of solid feed, bacteria such as *Prevotella* therefore gradually occupy the dominant position. Beyond this, other genera such as *Blautia* (7, 20, 33, 42), *Paraprevotella* (36, 42), *Oscillibacter* (42, 43), *Roseburia* (16, 20, 33, 36, 42), *Ruminococcus* (16, 20, 33, 36), *Oscillospira* (16, 33, 43), *Coprococcus* (20, 36), *Faecalibacterium* (20, 36), and *Treponema* (36) increase in abundance in the gut of piglets from newborn to weaning. However, genera like *Bacteroides* (16, 20, 33, 36, 42, 43), *Parabacteroides* (20, 36), *Fusobacterium* (7, 16, 20, 36, 42), *Lactobacillus* (16, 36, 42), *Anaerotruncus* (20, 33), *Butyricimonas* (20, 33), *Streptococcus* (16, 20, 36, 43), *Clostridium* (16), and *Escherichia* (33, 36, 42, 43) decline.

An unstable gut microbial community is one of the main causes of diarrhea in suckling piglets (44). Fluctuations within the gut microbiome, such as decreased abundances of Actinobacteria, Firmicutes (i.e., *Clostridium*, *Ruminococcus*, *Blautia*, and *Enterococcus*) (45), *Prevotella*, and *Lactobacillus* (34), and increased abundances of *Enterococcus* and *Escherichia coli* (46), have been identified in neonatal piglets with diarrhea and acute necrotizing enterocolitis (46, 47). The abundance of bacteria belonging to the phylum Fusobacteria in diarrheic piglets can be four times higher than in healthy piglets (45). Such disorders within the microbiome do not only increase the risk of diarrhea and the mortality of young pigs, but can also impact the absorption of nutrients and the anti-inflammatory regulation of the host.

**Development of gut microbiota in piglets from weaning to 7-days postweaning (21 to 28 days).** Weaning is the most stressful time in the piglets' life. During this period, piglets are switched from milk- to a solid-based diet (48–50). Accumulating

**FIG 1** Legend (Continued)

different growth stages. Principal coordinate analysis (PCoA) plots based on the Bray-Curtis distances show distinct clusters. The four growth stages are differentiated by colors: purple for lactation, blue for nursery, green for growing, and red for finishing. The origin of data is differentiated by dots in different colors: blue for United States and red for China. (C) The dynamic distribution of the 10 main bacterial genera across ages. C-1, *Escherichia*; C-2, *Lactobacillus*; C-3, *Prevotellaceae-Prevotella*; C-4, *Closridium*; C-5, *Bacteroides*; C-6, *Oscillospira*; C-7, Fusobacterium; C-8, *Paraprevotellaceae-Prevotella*; C-9, *Treponema*; C-10, *Bacillus*.

evidence indicates that piglets have an abrupt shift in the gut microbiota taxa and function during this transition period (51–53), leading to poor growth performance, or even death (54).

During the first week after weaning, the microbiota in the gut of healthy pigs changes dramatically. As mentioned above, an early colonization of specific bacterial groups (such as *Prevotella*, *Roseburia*, *Blautia*) in the GIT might offer piglets a stronger ability to digest glycan (55), which prepares the host to cope with a postweaning diet rich in complex carbohydrates. Compared with postweaning piglets with diarrhea, healthy piglets harbor a higher abundance of bacteria belonging to the *Prevotellaceae*, *Lachnospiraceae*, *Ruminocacaceae*, and *Lactobacillaceae*, even as early as 7 days after birth (34). Species belonging to the genera *Prevotella* and *Roseburia* are not only adapted to metabolize a wide range of complex carbohydrates, but also are major producers of short-chain fatty acids (SCFAs) (34, 56), affecting the energy metabolism and intestinal health of the host. *Prevotella* and *Roseburia* species are also reported to be negatively associated with *E. coli*-induced enteric infections (57).

Timely intake of solid feed is interestingly and strongly related to the abundance and proportion of *Prevotella* and *Bacteroides*. It is common that piglets do not eat feed in the immediate postweaning period due to weaning stress, and the abundance of *Prevotella* is temporarily decreased while *Bacteroides* is increased in abundance. For example, from day 21 (day of weaning) to day 24, the abundance of *Prevotella* (33.07%), *Alloprevotella* (12.81%), and *Oscillibacter* (5.52%) in the gut of piglets can decrease to 13.9%, 11.0%, and 4.8%, respectively. However, the abundance of *Bacteroides* (5.22%), *Faecalibacterium* (0.05%), and *Roseburia* (1.77%) can increase to 6.31%, 8.97%, and 3.51%, respectively (58). On the other hand, once the newly weaned piglets start to consume feed, the abundance of *Prevotella* can significantly increase from 12.93% (day 21) to 57.24% (day 28) (51, 55). *Prevotella* (23.1%) is one of the three most abundant genera identified in the feces of weaning piglets (21 days old), and it becomes predominant (34.7%) in the feces of post-weaning pigs (28 days old), followed by *Ruminococcus* (5.2%) and unclassified bacteria belonging to the *Lachnospiraceae* (5.0%) (33). Similarly, another study shows that the abundance of *Prevotella* and *Lactobacillus* is significantly higher in the gut of post-weaning piglets (28 days old) compared with weaned piglets (21 days old), accompanied by changes in the metabolism of carbohydrates and amino acids, which may be beneficial for the adaptation to the new diet post-weaning (55). On the day of weaning, the gut microbiota in healthy piglets is dominated by *Prevotella*, *Lactobacillus*, and *Fusobacterium*, which shift quickly into a community dominated by *Prevotella*, *Roseburia*, and *Clostridium* within 10 days (42). This relatively stable microbiota is efficient at degrading complex carbohydrates such as dietary fibers and thus is a "mature-like microbiota" to help healthy pigs better adapt to diets during the weaning transition period (42).

Weaning may trigger "unhealthy" alterations in the microbial composition, resulting in postweaning diarrhea (PWD), which is a major problem in the swine industry and causes significant economic losses (49). The diversity of the gut microbial community and composition during suckling period is believed to be associated with the susceptibility of pigs to PWD (34). One of the evident examples is the increase of opportunistic pathogens such as *Campylobacter* and the decrease of beneficial bacteria like *Alloprevotella* or *Oscillibacter* during the weaning transition period (58). In the GIT of piglets weaned at day 28 and challenged with *E. coli*, the relative abundance of *Turicibacter*, *Clostridium*, *Campylobacter*, *Dehalobacterium*, and *Desulfovibrio* are increased, while the abundance of *Paludibacter*, *Prevotella*, *Blautia*, *Faecalibacterium*, *Lactobacillus*, and *Coprococcus* are decreased (17). Enterobacteriaceae is a large family of Gram-negative bacteria with similar biological characteristics living in the GIT of humans and other mammals, including pigs (34, 59). Some bacteria belonging to the Enterobacteriaceae are the most important pathogens of intestinal infectious diseases (34). The enrichment of Enterobacteriaceae bacteria in the gut of pigs with diarrhea at day 30 can be observed 1 week prior to diarrhea symptoms, which means bacteria

belonging to the Enterobacteriaceae could be a biomarker for diarrhea in piglets during the weaning period (34). Few studies have indicated that the changes in the microbiota in the GIT of piglets with diarrhea may directly alter the intestinal function and metabolism of host (58). Five metabolic pathways including phenylalanine metabolism, citrate cycle (TCA cycle), glycolysis, or gluconeogenesis, propanoate metabolism, and nicotinate and nicotinamide metabolism were found to be involved in stress-induced microbial dysbiosis in the gut of weaning pigs (58).

**Development of gut microbiota in pigs from days 28 to 154.** As pigs age, the diversity and richness of microbiota in the gut increases (Fig. 1A; Table S2) (60). According to a large number of studies, *Prevotella* spp. are the most abundant bacteria in the GIT of pigs between 28 to 91 days (10, 13, 60–62), but it decreases from 91 to 154 days of age (10, 61). The proportion of *Anaerobacter* spp. increases as pigs grow, and it becomes the most abundant genus of bacteria in pigs at 154 days (61). With time, other bacteria such as *Lactobacillus*, *Fusobacterium*, *Oscillospira*, *Escherichia*, *Roseburia*, *Faecalibacterium*, and *Bacteroides* are also part of the dominant microbiota in the gut of pigs during grower-finisher period. During this period, the feed intake and the body weight of pigs increases dramatically. Such stable gut microbiota can reduce the risk of infectious intestinal diseases and ensure that the growth potential of animals can be brought into full play.

**Characteristics of GIT microbiota of healthy pigs at different ages revealed by meta-analysis.** Due to the physiological, nutritional, and immunological contributions of gut microbiota to the host, its structure and especially its function need to be assessed in more detail (63). On the other hand, recent studies have revealed that the structure of several microbial ecosystems can be effectively examined, which suggests that the microbiota of individuals can be clustered into so-called "enterotypes" based on the composition at the genus level (36, 41, 64–67). Although the composition of the gut microbiota in pigs at different ages has been investigated, the description of the characteristics of different microbes is still lacking. We have summarized that the microbiota in pig GIT dynamically changes with age in the previous sections, but it is still uncertain whether there are specific microbial groups in the GIT of pigs at different physiological stages.

Of 57,875,211 quality-filtered sequences from the feces of 1,192 healthy pigs without special treatment, 39,635,021 (68.5%) sequences showed at least 97% similarity to sequences in the SILVA database. The donors range from newborn (day 0) to finishing (day 174), helping to draw a full picture of the dynamic distribution of the microbiota at different ages. Finally, a total of 12,595 bacterial operational taxonomic units (OTUs) were identified and assigned to 39 phyla and 509 genera. Differences in the $\alpha$-diversity of microbial communities were observed among suckling, nursery, growing and finishing pigs, with finishing pigs harboring the most diverse microbiota (Fig. S2A to F). Principal coordinate analysis (PCoA) plots based on Bray-Curtis showed shifts in the microbial community among these four stages, especially before (lactation) and after weaning (nursery) phase (Fig. 1B). These results indicate that the weaning period may be a powerful window for regulating the gut microbiome of pigs.

We next tried to identify the "core" and "stage-associated" microbiome in the gut of healthy pigs. The definition of a "core microbiome" in the swine gut has been the focus of numerous publications (62, 68), which is intriguing as it may provide novel targets for dietary or therapeutic interventions. Based on the average relative abundance, microbial members with high abundance throughout the life can be accepted as part of the "core" microbiome in the swine gut (62, 68). On the other hand, microbial species that only appear in the GIT of pigs at specific growth stages are proposed as "stage-associated" microbiome (62).

Although a "core" microbiota may not exist in swine according to a strict definition, here we still found several bacterial phyla and genera in more than 90% of the samples in all growth stages. Firmicutes and Bacteroidetes accounted for the majority of total sequences (72.93% in lactating pigs, 89.32% in nursery pigs, 89.78% in growing pigs, and 87.42% in finishing pigs). A total of 19 genera were found in more than 90% of samples (Fig. S3,

4), which were defined as "core" bacteria here. Accordingly, *Bacteroides*, *Escherichia*, and *Lactobacillus* are predominant in lactating pigs (Fig. S3A), while *Prevotella*, *Lactobacillus*, and *Oscillispira* are dominant in nursery pigs (Fig. S3B). In growing pigs, the three most abundant genera are *Prevotella*, *Lactobacillus* and *Faecalibacterium* (Fig. S3C), and the top 3 genera are *Prevotella*, *Lactobacillus*, and *Streptococcus* in finishing pigs (Fig. S3D). These dominant taxa may be the potential functional microbiota, not only for their relatively high abundance, but also for their higher frequency to contact with the intestinal mucosa of host.

Using the linear discriminant analysis (LDA) and effect size (LEfSe), the "stage-associated core bacteria" were identified (Fig. S5, 6). In detail, *Bacteroides*, *Escherichia*, *Lactobacillus*, *Fusobacterium*, and *Ruminococcus* were identified as the lactation stage-associated genera. *Blautia* was identified as a nursery stage-associated genus. Growing stage-associated genera included *Prevotella*, *Megasphaera*, *Treponema*, and *Faecalibacterium*, while finishing stage-associated bacteria were characterized by *Streptococcus*, *Clostridium*, *Prevotella*, *YRC22*, and *SMB53*. It is worth noting that the abundance of bacteria belonging to the Proteobacteria (17.35%) showed higher relative abundance in suckling piglets compared with other growth phases (Fig. S7A; Table S4). Ten genera, including *Bacteroides*, *Escherichia*. and *Prevotella*, were present across all stages (Fig. 1C; Fig. S7B; Table S5). Although *Bacteroides* (15.59%, dominated by *B. fragilis*) and *Escherichia* (12.45%, dominated by *E. coli*) were present in the gut of suckling piglets, they were markedly decreased in the subsequent growth stages (Fig. S7B, C; Table S6). Bacteria belonging to the family Enterobacteriaceae, such as *E. coli*, often show over-growth in the case of dysbiosis (69) and may become the dominant bacteria, leading to the exacerbation of gut damage and diarrhea (46, 70). Previous studies have focused on the enrichment of Enterobacteriaceae in swine gut caused by dietary and environmental challenges, but our analysis provides another hypothesis: the distinct high abundance of *Escherichia* in the GIT of suckling pigs may contribute to the susceptibility of dysbacteriosis associated diarrhea. The high abundance of *Bacteroides* during pre-weaning may be due to their ability to utilize monosaccharides and oligosaccharides present in sows milk (36). Notably, *B. fragilis* is a symbiont found in the colon and colonizes mucus or epithelium (71). It can protect the host from multiple preclinical colitis via the induction of the anti-inflammatory response, such as increased interleukin-10 production by Foxp$^{3+}$ regulatory T cells (72, 73). In addition, a dramatic increase of *Prevotella* was found postweaning, and was the most abundant genera in the gut of nursery (22.32%), growing (27.79%), and finishing (15.25%) pigs (Fig. S3B to D; Fig. S4B; Table S6), suggesting that *Prevotella*, a class of bacteria with powerful capacity to metabolize complex carbohydrates like hemicelluloses and xylans (40, 41, 51), may be the dominant symbiont in the GIT of healthy post-weaning pigs.

The term "enterotype" is now recognized as an important characteristic of the gut microbiome. A total of three enterotypes, E1 (*Bacteroides-Escherichia-Lactobacillus*), E2 (*Prevotella-Lactobacillus-Megasphaera*) ,and E3 (*Prevotella-Lactobacillus-Treponema*), were identified from the collected sequences (Fig. S8 to 9). Among them, E1 accounted for approximately 85% of the bacterial genera in the gut of suckling piglets (Fig. S9A), while E3 accounted for 88% in finishing pigs (Fig. S9D). For nursery and growing pigs, the division of enterotypes was not absolute. The proportion of E1, E2, and E3 in the gut of nursery piglets was 16%, 50%, and 34%, respectively (Fig. S9B). However, only two enterotypes, E2 (60%) and E3 (40%), were found in the gut of growing pigs (Fig. S9C). *Bacteroides*, *Escherichia*, and *Lactobacillus* were the three main genera of bacteria contributing to E1 (Fig. S10A, B, and D). *Megasphaera*, *Prevotella*, and *Lactobacillus* were the other three main genera contributing to E2 (Fig. S10B, C, and E), while *Prevotella*, *Treponema*, and *Lactobacillus* were the main genera contributing to E3 (Fig. S10B, E, and F). These findings can help to understand the whole picture of how enterotypes change from suckling to finishing pigs, which can be one of the characteristics used to assess balanced or normal microbiota in healthy pigs at different growth stages.

We further determined which variables (age, study, weaning day, creep feed, growth stages, enterotype, origin, and sequencing platform) most strongly affected

the composition of the swine feces microbiota (Table S7), and the result indicate that age is the most important factor affecting the gut microbiota of pigs. This aligns with our previous summary that the microbial community dynamically changes with age.

There are several reviews that shown more or fewer similarity with ours, but the focus is totally different. Patil et al. and Aluthge et al. summarize the different factors (i.e., age, birth, breed, and diet) influencing the pig gut microbiota, and focus on how commensal microorganisms impact the biochemical and metabolic process of host (74, 75). Different from this, Gresse et al. focus on the postweaning diarrhea caused by gut microbial dysbiosis, as well as the *in vitro* models of piglet gut for the development and testing of new feed additives (52). Here, we do not only summarize the microbial composition in swine gut of different ages, but also try to screen some potential functional microbiota. Further research targeting these "core" microbes and the dynamic distribution of microbiota, as well as the related function is of great importance in swine production.

**The microbial load along with pig GIT or among individuals.** In a recently published meta-analysis, the microbial load in the GIT and the dynamic shifts of the microbiota along with the GIT of pigs were discussed (68). A clear demarcation between the microbiota in the digestion samples from upper and lower GIT was found, with significantly higher diversity, richness, and evenness in the samples from the lower GIT (68). This finding is further supported by several later studies from 2017 to 2020, which confirm strong structural and functional differences in the colonized microbial populations between the upper and lower GIT (8, 11, 76, 77). At the same time, a similar difference in microbial load is also found among different individuals. A human study showed a 10-fold difference in the microbial load among healthy individuals, which may be a key driver of observed microbiota alterations in patients with Crohn's disease (78). Some phenotypes, such as the moisture in feces, are also proved to be linked to fluctuations of microbial load. A revisit of a disease association microbiome data set, comprising 106 patients with primary sclerosing cholangitis and/or inflammatory bowel disease, shows a negative correlation between the microbial loads and intestinal and systemic inflammation markers (79). However, similar analysis or studies on the interaction between the microbial loads and phenotypes of pigs are lacking. As the functional output of the microbiome is very different and depends on the microbial loads, this can be a novel and important topic in microbiome related studies in pigs.

**Contribution of the gut microbiota to host phenotypes.** Due to the rapid development of sequencing techniques, data are available to link certain gut bacteria with swine phenotypes. The identification of specific bacterial biomarkers linked to growth performance in a supervised way will provide promising information to develop effective strategies for modulating the gut microbiota and improving growth performance of pigs (36, 80).

**Body weight and the gut microbiota.** The association between growth performance and the gut microbiota in pigs has been the center of discussion in recent years. Birth weight has been reported to strongly influence the bacterial composition in the GIT of piglets from days 7 to 21 (43). A relationship is also found between the body weight and microbial composition in swine gut. For example, the levels of *Bacteroides* (2.65% versus 4.54%), *Anaerotruncus* (0.01% versus 0.03%), and *Anaerococcus* (0.02% versus 0.01%) were significantly different in the gut of the heavier (16.70 to 22.75 kg) pigs compared with the lighter (8.09 to 11.89 kg) pigs (81). The body weight of finishing pigs at 136 days of age was also observed to be positively correlated with the abundance of Firmicutes and negatively correlated with Bacteroidetes (82).

Studies on the characteristics of each enterotype show that there is an indicative bacterial group in the center of the commensal microbial network, suggesting that specific enterotype may be closely related to certain phenotypes of host (36, 66). However, interactions between enterotype and phenotype are rarely studied. Limited references indicate that piglets with *Prevotella*-dominated enterotypes present lower growth rate during the lactation period, but higher body weight and average daily gain after weaning (36).

**Food intake and the gut microbiota.** Growing evidence suggests that the gut microbiota may play an important role in the regulation of appetite and feeding behavior of pigs. A correlation has been found between enterotypes and food intake of growing-finishing pigs (66). For instance, pigs with *Prevotella*-dominated enterotype have higher food intake than those with *Treponema*-dominated enterotype (66). Specifically, a total of 12 OTUs assigned to *Prevotella* and six OTUs assigned to Lachnospiraceae, *Faecalibacterium prausnitzii*, Ruminococcaceae, *S24-7*, *Anaeroplasma*, and *Sutterella* are found to be positively related to food intake (66). This indicates that *Prevotella* may be a key genus and a potential new target for increasing the food intake of pigs. How the gut microbiota affects the appetite of pigs remains unknown, but research using other animal models may provide useful references (83, 84). The pathways associated with the satiety of the host can be activated by some bacteria-derived proteins, and the control of ingestion by the host may be influenced by the growth cycle of bacteria (83). For example, *E. coli* or other species affiliated with Enterobacteriaceae have been reported to synthesize mimetic proteins of peptide hormones, which may influence the appetite of host via the activation of anorexigenic pathways (83). A positive correlation was also observed between the abundance of *Prevotella* and serum ghrelin, the only appetizing hormone known at present (84). However, whether the microbiota in pig GIT influences the appetite of host remains unclear, and the causality between the change in gut microorganisms and feeding behavior of host also needs to be revealed.

**Feed efficiency and the gut microbiota.** Feed efficiency (FE, higher is better) is another important economic metric in addition to body weight and feed intake in swine production. Different compositional and functional characteristics of gut microbiota have also been reported to be associated with FE in pigs. Taking *Prevotella* as an example, although it is the predominant genus in the cecum of 166-day old pigs with high (FCR = 2.03 $\pm$ 0.04, lower is better) and low FE (FCR = 2.85 $\pm$ 0.19), a remarkably high abundance of *Prevotella* sp. CAG:604 was only found in the gut of low-FE pigs, indicating the potential of *Prevotella* as a biomarker to distinguish the cecal microbiota of pigs with high and low FE (85). The linkage between the gut microbiota and FE is also supported by another study focusing on the fecal/intestinal (ileal and cecal) bacterial profiles and residual feed intake (RFI, lower is better) throughout the lifetime of pigs (86). Of the 13 bacterial genera correlated with RFI, eight genera are negatively correlated with lower RFI ($-51.0 \pm 15.40$) and five genera are positively correlated with higher RFI ($76.0 \pm 15.40$) across different sample types. Among them, *Butyrivibrio* (42 days postweaning) and *Prevotella* (at weaning) are the two genera positively correlated with a low RFI value, which may be due to the introduction of cereal-based diet rich in complex carbohydrates post-weaning (86). An OTU-based association analysis shows that 31 OTUs assigned to bacteria related to the metabolism of dietary polysaccharides are potentially linked to FE in 140-day old finisher pigs (64). Most of these "RFI-associated" bacterial OTUs are related to the orders *Clostridiales* and *Bacteroidales*, the families *Ruminococcaceae*, *Christensenellaceae*, and *Lachnospiraceae*, as well as the genera *Prevotella* and *Faecalibacterium*. The abundance of bacterial OTUs in ileum (11), cecum (55), and colon (55) are also reported to be different between 140-day old pigs with low FE (FCR = 2.65 $\pm$ 0.07) and high FE (FCR = 2.23 $\pm$ 0.07) (87). All these findings indicate that regulating the microbial composition in swine gut may be conducive to developing strategies to improve FE. The distinctive microbiota in the GIT of pigs with high FE may be more competent in terms of digesting dietary carbohydrates (64, 85–88). Functional analysis reveals that pathways associated with pyruvate-related metabolism are over presented in pigs with higher FE (FCR = 2.03 $\pm$ 0.04) than those with lower FE (FCR = 2.85 $\pm$ 0.19) (85). Pathways related to amino acids, cancers, signaling molecules and interaction, metabolism of cofactors and vitamins, digestive system, glycan biosynthesis and metabolism, and immune system are also observed to be more abundant in the colon of 140-day old pigs with high FE (FCR = 2.23 $\pm$ 0.07) than those with low FE (FCR = 2.65 $\pm$ 0.07) (87). KEGG database orthologs related to nitrogen metabolism and transport system, amino acid metabolism and transport system, glycine,

serine, and threonine metabolism are positively associated with the FE of 140-day finishing pigs (64). Such findings improve our understanding of how the gut microbiome influences porcine FE. Characterizing "FE-associated" microbial biomarkers may help to define an "optimal" microbial profile to improve the FE and growth performance of pigs.

The cross talk between gut microbiota and host phenotype is very complex as gut microbiota can be both a cause and result. Here, we try to summarize the relationship between some target microbes and the host phenotypes, which will be the foundation of further research: whether it is possible to achieve target growth performance by regulating this microbiota (a cause), or as a biomarker to indicate the target growth performance or health status of host (a result).

The current work extends the understanding of the dynamic shift of gut microbes throughout each growth stage of pigs (Fig. 2). It also characterizes the profiles of bacterial communities across pig GIT, which helps to define an "optimal" microbial profile or specific microbes with beneficial functionality. Our comprehensive summary and meta-analysis confirm significant dynamic shifts in the gut microbiota of pigs at different growth phases. In general, the gut microbiota of suckling pigs may be dominated by *Bacteroides*, *Escherichia*, *Clostridium*, *Lactobacillus*, *Fusobacterium*, and *Prevotella*, followed by age-associated shifts to a microbiota with *Prevotella* and *Aneriacter* as predominant genera and subsidiary genera, such as *Fusobacterium* and *Lactobacillus*. Postweaning diarrhea is a common issue in swine production which is traditionally controlled by ZnO or antibiotics in feed. However, the ban on feed antibiotics by more and more countries and the expected ban on ZnO in feed supplementation from 2022 in the EU urge researchers and pig producers to search for new alternatives. One possible alternative is to use novel probiotics, such as the so-called "next-generation probiotics (NGPs)" derived from GIT. We reveal that a total of 19 "core" bacterial genera dominated by *Bacteroides*, *Prevotella*, and *Lactobacillus* can be found in more than 90% of healthy pigs across different ages, and three enterotypes can also be identified. Without exception, both the composition of these "core" bacteria and enterotypes are gradually changed into an "adult-like" profile with age. These identified genera may probably be the potential candidates of NGPs or the potential target of microflora regulation. New evidence proves that some of these "core" microorganisms like *Lactobacillus*, *Prevotella*, *Bacteroidetes*, and *Fusobacteria* contribute to the growth phenotypes of pigs. Therefore, some consensus, such as the FCR of finisher pigs is much higher than piglets, or that piglets are prone to stress-induced diarrhea, may be partially due to their unique gut microbiota, in which the contribution of specific bacteria (i.e., abundant *Prevotella*, *Escherichia*, or other contributors of enterotypes) cannot be ignored. Based on this, a precise nutritional strategy for pigs may have to be considered, along with the requirements of the host and the target microorganisms (such as the "core" microbiota we have screened). Adding substrates preferred by these target microbes will help to increase the abundance of specific symbiotic species and benefit the gut health of pigs. Although we have comprehensively summarized the dynamic shifts of gut microbiota in pigs, substantial work remains to investigate metabolites as the functional output of microorganisms to interact with the host phenotype. It is also possible to achieve the desired phenotype through effective regulation of constantly changing microbial communities or metabolites. Further research targeting these "core" microbes and the dynamic distribution of microbiota, as well as the related function is of great importance in swine production.

## MATERIALS AND METHODS

**Data acquisition and study inclusion criteria.** The meta-analysis was carried out following the reference of Holman et al. (68). Studies which included in the meta-analysis were described in Table S3. Studies were identified through a literature search of NCBI PubMed and Google Scholar as far as 2020. The search strategy also refers to Holman et al. (68). Sequence files from each study were downloaded from the SRA or European Nucleotide Archive (ENA; http://www.ebi.ac.uk/ena).

**Processing of 16S rRNA gene sequences.** All downloaded 16S rRNA gene sequences were processed using the QIIME software package (v. 1.9.1) (89). The maximum length of each read was set based

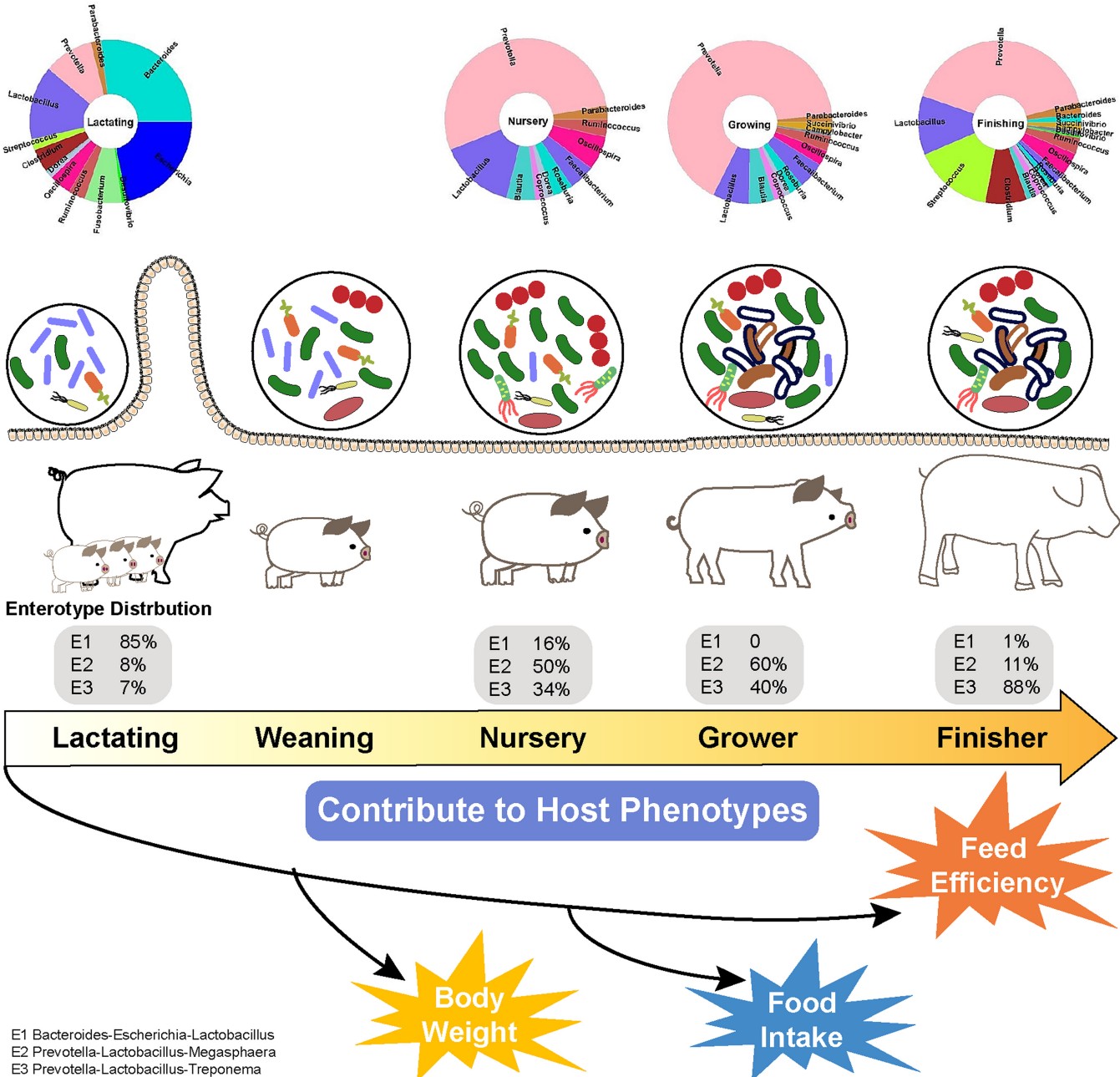

**FIG 2** A system biology model for the dynamic distribution of the gut microbiota in pigs' life: composition and contribution. The gut microbiota in pigs changes remarkably across different ages. A total of 19 so-called "core" bacterial genera leaded by *Bacteroides*, *Prevotella*, and *Lactobacillus* can be found in more than 90% of healthy pigs at different ages, and three enterotypes can also be identified.

on the expected read length, as reads spanning regions V1 to V6 region. The quality-filtered 16S rRNA gene sequences were clustered into OTUs at 97% similarity using a closed-reference approach. Of 57,875,211 quality-filtered sequences from fecal samples of 1,192 healthy pigs without special treatment, 39,635,021 (68.5%) sequences showed at least 97% similarity to sequences in the SILVA database (90). All samples were randomly sub-sampled to a level of 1,000 sequences per sample to account for uneven sequencing depth among samples and studies.

Samples were categorized based on treatment, sample choice, number of samples, sample type, DNA extraction method, country of origin, age, sequencing platform, and 16S rRNA hypervariable region sequenced, where available. Linear discriminant analysis effect size (LEfSe) was used to determine which genera were significantly enriched in each sample type. Genera that were relatively more abundant in a particular sample group were identified by LEfSe using the Kruskal-Wallis test ($P < 0.05$), and the effect size of each of these genera was estimated using linear discriminant analysis (91). An LDA score ($\log_{10}$) of 4.0 was used as the cutoff for identifying differentially abundant genera. The between-sample (beta)

diversity was assessed using the unweighted and weighted UniFrac distances and Bray-Curtis dissimilarities (92, 93). PCoA was used to visualize these distances using Emperor (94). Permutational multivariate analysis of variance (PERMANOVA) using the adonis function with 9,999 permutations was implemented in QIIME to analyze the unweighted and weighted UniFrac distances and the Bray-Curtis dissimilarities for each gastrointestinal location, country of origin, hypervariable region, sequencing platform used, and study. The within-sample (alpha) diversity, richness, and evenness were calculated within QIIME using the Shannon index, phylogenetic diversity (PD whole tree), Simpson reciprocal index, and equitability (evenness) index. These metrics were compared among pig age using a two-way ANOVA in R (v. 3.2.5) (95) with hypervariable region and sample type as the independent factors, followed by Tukey's honestly significant difference (HSD) *post hoc* pairwise comparison test (agricolae package) (96). All results were considered significant at $P$ values of $<0.05$.

## SUPPLEMENTAL MATERIAL

Supplemental material is available online only.
**SUPPLEMENTAL FILE 1**, PDF file, 1.9 MB.

## ACKNOWLEDGMENTS

The work presented in this manuscript was supported by the National Natural Science Foundation of China (NSFC, Grant numbers 31730091, 31872369, 32072743, and 31672436).

Y.L. and W.R. obtained the sequence data, and designed and wrote the manuscript; U.M.M. and A.J.C. conceived the meta-analysis; H.S., A.G.W., G.S., and R.I.M. revised the manuscript and provided critical feedback; B.Y. and J.Y. helped with designing figures; J.H. and H.Y. helped with designing tables and inputting data; J.W. helped with finding relevant literatures; D.C. approved the final manuscript.

All the authors read and approved the final version of the manuscript. We declare no competing interests.

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
