## [Reviewer comments · Microbiology Spectrum]

Microbiology Spectrum

Dynamic distribution of gut microbiota in pigs at different growth stages: composition and contribution

Yuheng Luo, Wen Ren, Hauke Smidt, André-Denis Wright, Bing Yu, Ghislain Schyns, Ursula McCormack, Aaron Cowieson, Jie Yu, Jun He, Hui Yan, Jinlong Wu, Roderick Mackie, and Daiwen Chen

Corresponding Author(s): Daiwen Chen, Institute of Animal Nutrition, Sichuan Agricultural University, Chengdu, Sichuan

Review Timeline:

Submission Date:	June 23, 2021
Editorial Decision:	October 9, 2021
Revision Received:	December 6, 2021
Editorial Decision:	January 19, 2022
Revision Received:	February 26, 2022
Accepted:	March 6, 2022

Editor: Cheng-Yuan Kao

Reviewer(s): Disclosure of reviewer identity is with reference to reviewer comments included in decision letter(s). The following individuals involved in review of your submission have agreed to reveal their identity: Rustem Abuzarovich Ilyasov (Reviewer #3)

Transaction Report:

DOI: <https://doi.org/10.1128/spectrum.00688-21>

October 9, 2021

Prof. Dai wen Chen
Institute of Animal Nutrition, Sichuan Agricultural University, Chengdu, Sichuan
chengdu
China

Re: Spectrum00688-21 (Dynamic distribution of gut microbiota in pigs at different growth stages: structure, contribution, and regulation)

Dear Prof. Dai wen Chen:

Thank you for submitting your manuscript to Microbiology Spectrum. When submitting the revised version of your paper, please provide (1) point-by-point responses to the issues raised by the reviewers as file type "Response to Reviewers," not in your cover letter, and (2) a PDF file that indicates the changes from the original submission (by highlighting or underlining the changes) as file type "Marked Up Manuscript - For Review Only". Please use this link to submit your revised manuscript - we strongly recommend that you submit your paper within the next 60 days or reach out to me. Detailed information on submitting your revised paper are below.

Link Not Available

Sincerely,

Cheng-Yuan Kao

Journals Department
Reviewer comments:

Reviewer #1 (Comments for the Author):

This review manuscript summarized the characteristics and the dynamic shifts in the gut microbial community of pigs at different ages based on the results of 63 peer-review publications, and further meta-analyzed the sequences from 16 studies with accession numbers in the GenBank database. The review manuscript is lack of comprehensive discussion and does not provide a new insight for readers. The most drawback is the meta-analysis of the sequences from other papers.

1. Lots of parameters affect the pig microbiota, besides difference stages, such as breed, genetics, nutrition, feed additives (supplements), antibiotics and environment (geography). Other factors, such as the sampling approach, extraction methods, and sequencing conditions, are also influence the outcomes of microbiota. It is very difficult to combine the sequencing data from different papers under many diversities of background conditions, which explains why only dominant taxa were found without new Interpreting the findings at the different growth stages.
2. The topic here "Dynamic distribution of gut microbiota in pigs at different growth stages: structure, contribution, and regulation" seems not match the content of manuscript. I suggest that the problems to be addressed by the review should be specified in the form of clear, unambiguous and structured questions before beginning the review work.
3. Several recent papers have provided comprehensive reviews related to gut microbiota and pigs. Most information in the

present manuscript can be found in following papers. The present manuscript does not provide a new insight and is also lack of deep discussion regarding this topic.

Y Patil, R Gooneratne, XH Ju. Interactions between host and gut microbiota in domestic pigs: a review- Gut Microbes, 2020.

ND Aluthge, DM Van Sambeek et al. BOARD INVITED REVIEW: The pig microbiota and the potential for harnessing the power of the microbiome to improve growth and health- Journal of animal Science. 2019.

Raphaële Gresse,Frédérique Chaucheyras-Durand, et al. Gut Microbiota Dysbiosis in Postweaning Piglets: Understanding the Keys to Health. Trends in Microbiology. 2017.

Reviewer #2 (Comments for the Author):

The review by Luo and Ren et al describes in detail a meta-analysis geared towards identifying patterns in the porcine gut microbiota. The manuscript is topical, well-written, the methods are adequate and the results are well detailed.

A few comments that would improve the manuscript:

- The main question here is well known by everybody working in the microbiome area. Are the microbiome changes a cause or an effect? The authors need to include statements that qualify this, if possible...
- The authors also need to provide some practical advice, since there is a danger of well-prepared manuscripts such as this one to be highly theoretical. The one question I would like addressed here is what would be the main practical outcome of this manuscript? If this would have to be boiled down to one paragraph that would help a pig producer to get a better value (and better welfare) for his pigs, what would that paragraph be?
- My view is that resilience of the microbiome (and the animal as a whole) is the one aim of pig production. Can the authors comment on this from this meta-analyses? I agree that resilience can mostly be assessed through a side-by-side comparison and a change in some experimental factors...
- Are there details of medicated feed available in any of the studies reviewed here, since medicated and even zinc oxide feed would have an effect on the microbiome. And in light of the 2022 changes to the zinc oxide legislation in Europe, are these results applicable to the post zinc oxide ban?

Specific comments

L69 - I would like to see the materials and methods here, not in the supporting material

L69 - please include an age threshold for each production stage (pre-weaning-nursery-growth-finish)

L76 - please clarify is the nursery is considered the post-weaning period

L88 - can the stability of the microbiome be calculated?

L137 - insert anti-inflammatory instead of ant-inflammatory

L153 - this may be of use when discussing Prevotella Intestinal microbiota of pigs determines the response to vaccines - Articles - pig333, pig to pork community

L174 - does adult like mean? stable?

L200 - how do you objectively assess stability?

L225 - not sure about the intricate details of the meta-analysis, but sometimes, an age-related analysis can be distorted by extreme age values. For example if you only have a few pigs that are 170-day old pigs (compared to many more at weaning-nursery), any specific bacteria for those pigs are at a very low level, but not due to the fact that the bacteria are indeed low at that age, but because there is only a few pigs represented. Just wanted to alert the authors to this aspect...

L249 - replace top3 with top 3

L267 - please clarify the hypothesis - it would make sense that Escherichia are contributors to early life diarrhoeas, what is the novelty outlined here?

L319 - cause and effect issue here, if the authors have any comments on discerning whether the microbiome changes are a cause of an effect, they should be highlighted there

L362 - for this paragraph, the authors need to explain the relationship between feed efficiency (higher is better), FCR (lower is better) and RFI (lower is better), to clarify the data that are explained in the paragraph

L399 - as outlined above, can the authors include some practical recommendation for pig producers? What would make up a good/stable/resilient microbiome? Feed, add/remove medication, additives, appropriate care, a mix of these factors? In my view, if producers really care for their animals, they will make sure, they are looked after, they will offer them good quality feed, good quality accommodation, good healthcare and then the microbiome will follow...any opinions?

Staff Comments:

Preparing Revision Guidelines

Please return the manuscript within 60 days; if you cannot complete the modification within this time period, please contact me. If you do not wish to modify the manuscript and prefer to submit it to another journal, please notify me of your decision immediately so that the manuscript may be formally withdrawn from consideration by Microbiology Spectrum.

Dear editors and reviewers,

Thank you for giving us an opportunity to revise our manuscript. We appreciate for your and all the reviewers' comments and suggestions concerning our manuscript. All the comments are valuable and helpful for revising and improving our paper. Following those comments, we carefully revised the manuscript and responded to each review comment. Revised portions are highlighted in the "Marked-up Manuscript". At the same time, the point-by-point revisions to the comments and suggestions are listed as follows.

Reviewer #1 (Comments for the Author):

This review manuscript summarized the characteristics and the dynamic shifts in the gut microbial community of pigs at different ages based on the results of 63 peer-review publications, and further meta-analyzed the sequences from 16 studies with accession numbers in the GenBank database. The review manuscript is lack of comprehensive discussion and does not provide a new insight for readers. The most drawback is the meta-analysis of the sequences from other papers.

Response: thanks a lot for your careful reading and your comments. We totally understand your query, but we believe our review are meaningful to both swine research and swine production industry, especially for the research on alternatives of antibiotics. Even though there are more and more research on swine gut microbiota, the dynamic distribution and contribution of a healthy microbial ecosystem in pig gastrointestinal tract have yet to be qualitatively or quantitatively defined as a tool to help maximize animal health and growth performance. We only focus on the fecal microbiota in pigs with different age. It should be emphasized that these animals included in our analysis are health, natural growth and without any antibiotics or feed additives treatment. The meta-analysis based on existing studies will help to draw a map showing fecal microbiota composition in healthy pigs at different age.

Although your denial is very regrettable to us, we still would like to make a simple explanation of our review to show its importance. After all, a number of 19 bacterial genera, including *Bacteroides*, *Prevotella* and *Lactobacillus* can be found in more than 90% of pigs and three enterotypes can be identified in all pigs at different ages, suggesting there is a “core” microbiota in the gut of healthy pigs, which can be a potential target for nutrition- or health-regulation. The “core” members benefit the growth and gut health of the host. These findings help to define an “optimal” gut microbial profile for assessing, or improving, the performance and health status of pigs at different growth stages. Our review here is try to further summarize the relationship between some target microbiome and the host phenotypes, this will be the foundation of further research: whether it’s possible to achieve target growth performance by regulating these microbiota (a cause), or as a biomarker to indicate the target growth performance or health status (an effect).

Here, we carefully read and consider all your questions and suggestions, and we also humbly accept each of your query that helps us better improve our manuscript. We hope our efforts could receive your positive response. Thank you again for your comments on our manuscript.

1. Lots of parameters affect the pig microbiota, besides difference stages, such as breed, genetics, nutrition. feed additives (supplements), antibiotics and environment (geography). Other factors, such as the sampling approach, extraction methods, and sequencing conditions, are also influence the outcomes of microbiota. It is very difficult to combine the sequencing data from different papers under many diversities of background conditions, which explains why only dominant taxa were found without new Interpreting the findings at the different growth stages.

Response: thanks a lot for your comments. We totally agree with your opinion. Therefore, we have considered the influence of these parameters before the meta-analysis. We only collected and selected those microbial sequences under

limitations. In another word, only the data from so-called “normal pigs” (health, natural growth and without any antibiotics or feed additives treatment), which usually presented in the control treatment, or in those studies on the gut/fecal microbiota in healthy pigs with different age were included. For a more detailed understanding of the data included in our analysis, you can refer to Table S3 (partially shown as the photo below) where you can find that we have minimized the variations of influencing factors.

Table S3. Details of studies included in the meta-analysis⁴²

Author ⁴²	Treatment ⁴²	Sample choose ⁴²	Creep feed ⁴²	No. of samples ⁴²				Sample Type ⁴²	Hypervariable regions ⁴²	DNA extraction method ⁴²	Country of origin ⁴²	Sequencing platform ⁴²	Accession number for BioProject ⁴²
				Lactation (0-20 d) ⁴²	Nursery (21-60 d) ⁴²	Growing (61-116 d) ⁴²	Finishing (117-174 d) ⁴²						
Yang et al., 2017 ⁴²	Diarrhea or health piglet ⁴²	Health piglet ⁴²	Yes ⁴²	12 ⁴²	12 ⁴²	4 ⁴²	4 ⁴²	feces ⁴²	V4 ⁴²	TIANGEN Stool DNA Kit ⁴²	China ⁴²	Illumina HiSeq 2500 ⁴²	PRJNA340296 ⁴²
Han et al., 2019 ⁴²	Diarrhea or health piglet ⁴²	Health piglet ⁴²	NA ⁴²	24 ⁴²	4 ⁴²	4 ⁴²	4 ⁴²	feces ⁴²	V3-V4 ⁴²	TIANGEN DNA extract kit ⁴²	China ⁴²	Illumina MiSeq ⁴²	PRJNA450515 ⁴²
Poulsen et al., 2018 ⁴²	Bacillus or antibiotics ⁴²	NC ⁴²	No ⁴²	18 ⁴²	36 ⁴²	4 ⁴²	4 ⁴²	feces ⁴²	V1-V3 ⁴²	E.Z.N.A. stool DNA Kit ⁴²	Denmark ⁴²	Illumina MiSeq ⁴²	PRJNA503676 ⁴²
Li et al., 2018 ⁴²	Birth weight ⁴²	Normal birth weight ⁴²	Yes ⁴²	18 ⁴²	12 ⁴²	4 ⁴²	4 ⁴²	feces ⁴²	V3-V4 ⁴²	QIAamp Fast DNA Stool Mini Kit ⁴²	China ⁴²	Illumina HiSeq 2500 ⁴²	PRJNA448808 ⁴²
Chen et al., 2017 ⁴²	Age ⁴²	Different age ⁴²	NA ⁴²	17 ⁴²	51 ⁴²	4 ⁴²	4 ⁴²	feces ⁴²	V3-V4 ⁴²	Godon 1997 ⁴²	China ⁴²	Illumina MiSeq ⁴²	PRJNA381010 ⁴²
Guevarra et al., 2018 ⁴²	Age ⁴²	Different age ⁴²	NA ⁴²	4 ⁴²	20 ⁴²	4 ⁴²	4 ⁴²	feces ⁴²	V5-V6 ⁴²	QIAamp Fast DNA Stool Mini Kit ⁴²	South Korea ⁴²	Illumina Hi-Seq 2000 ⁴²	PRJNA437010 ⁴²

For those influencing factors which are not possible to minimize, we conduct the weighted and unweighted UniFrac distances and Bray-Curtis and Binary-Jaccard dissimilarities analysis (Table S7). We can learn from this analysis that age/growth stage of pigs is the most powerful parameter (indicated by the highest R² value) that influences the gut microbiota in pigs.

Table S7. Factors associated with the community structure of the swine gut microbiota as measured using weighted and unweighted UniFrac distances and Bray-Curtis and Binary-Jaccard dissimilarities⁴²

Parameter ⁴²	Value ⁴²											
	Unweighted UniFrac ⁴²			Weighted UniFrac ⁴²			Bray-Curtis ⁴²			Binary-Jaccard ⁴²		
	Pseudo-F ratio ⁴²	R ² ⁴²	P value ⁴²	Pseudo-F ratio ⁴²	R ² ⁴²	P value ⁴²	Pseudo-F ratio ⁴²	R ² ⁴²	P value ⁴²	Pseudo-F ratio ⁴²	R ² ⁴²	P value ⁴²
Age ⁴²	6.026 ⁴²	0.248 ⁴²	0.001 ⁴²	10.540 ⁴²	0.365 ⁴²	0.001 ⁴²	6.054 ⁴²	0.249 ⁴²	0.001 ⁴²	5.050 ⁴²	0.216 ⁴²	0.001 ⁴²
Study ⁴²	35.121 ⁴²	0.193 ⁴²	0.001 ⁴²	69.650 ⁴²	0.322 ⁴²	0.001 ⁴²	40.101 ⁴²	0.214 ⁴²	0.001 ⁴²	29.022 ⁴²	0.165 ⁴²	0.001 ⁴²
Weaning day ⁴²	31.415 ⁴²	0.176 ⁴²	0.001 ⁴²	60.143 ⁴²	0.290 ⁴²	0.001 ⁴²	34.872 ⁴²	0.192 ⁴²	0.001 ⁴²	26.232 ⁴²	0.151 ⁴²	0.001 ⁴²
Creep feed ⁴²	28.059 ⁴²	0.160 ⁴²	0.001 ⁴²	49.917 ⁴²	0.254 ⁴²	0.001 ⁴²	33.513 ⁴²	0.186 ⁴²	0.001 ⁴²	24.290 ⁴²	0.142 ⁴²	0.001 ⁴²
Growth stages ⁴²	11.095 ⁴²	0.070 ⁴²	0.001 ⁴²	18.634 ⁴²	0.113 ⁴²	0.001 ⁴²	11.943 ⁴²	0.075 ⁴²	0.001 ⁴²	9.607 ⁴²	0.061 ⁴²	0.001 ⁴²
Enterotype ⁴²	29.318 ⁴²	0.062 ⁴²	0.001 ⁴²	56.655 ⁴²	0.113 ⁴²	0.001 ⁴²	28.077 ⁴²	0.060 ⁴²	0.001 ⁴²	23.742 ⁴²	0.051 ⁴²	0.001 ⁴²
Origin ⁴²	2.524 ⁴²	0.006 ⁴²	0.021 ⁴²	0.493 ⁴²	0.001 ⁴²	0.461 ⁴²	5.916 ⁴²	0.013 ⁴²	0.001 ⁴²	3.601 ⁴²	0.008 ⁴²	0.001 ⁴²
Platform ⁴²	2.524 ⁴²	0.006 ⁴²	0.035 ⁴²	0.493 ⁴²	0.001 ⁴²	0.476 ⁴²	5.916 ⁴²	0.013 ⁴²	0.001 ⁴²	3.601 ⁴²	0.008 ⁴²	0.001 ⁴²

The new interpreting findings at different growth stages are important, but our review is also intended to summarize the microbial composition in pigs at

different ages and the potential functional microbiota to the growth performance of host. In this case, the dominant taxa maybe the potential functional microbiota, not only for their relatively higher abundance, but also for their higher frequency to contact with the intestinal mucosa of host. As indicated in our manuscript, a number of 19 bacterial genera, including *Bacteroides*, *Prevotella* and *Lactobacillus* can be found in more than 90% of pigs and three enterotypes can be identified in all pigs at different ages, suggesting there is a “core” microbiota in the gut of healthy pigs. These are the most highlights in our review and lay the foundation for further research on finding commensal microbiota benefit pig growth performance and gut health. In human beings, there have been researches showing the benefits of these dominant taxa, such as *Bacteroides*, *Prevotella*, et al. We can find some of the results in the following references (not completely listed):

- Zafar H., et al. Gut *Bacteroides* species in health and disease. Gut Microbes. 2021.
- Tett A., et al. *Prevotella* diversity, niches and interactions with the human host. Nature Reviews. 2021.
- Jacobson A., et al. A gut commensal-produced metabolite mediates colonization resistance to *Salmonella* infection. Cell Host&Microbe. 2018

We sincerely hope that our explain here is clear enough, which should earn your support on our manuscript. Thanks again.

2. The topic here "Dynamic distribution of gut microbiota in pigs at different growth stages: structure, contribution, and regulation" seems not match the content of manuscript. I suggest that the problems to be addressed by the review should be specified in the form of clear, unambiguous and structured questions before beginning the review work.

Response: thanks a lot for your comments. After carefully considering your suggestion and reviewing the content of our manuscript, we have revised the title

to “Dynamic distribution of gut microbiota in pigs at different growth stages: composition and contribution”. Because in our review, there are two main aspects: the composition of gut microbiota (represented by fecal microflora) in pigs with different age, and the contribution of gut microbiota to pigs’ phenotypes. However, a third part, ways to promote host performance by modulating the gut microbiota using feed additives, was also included in our first draft. This part was deleted in the submitted version due to the word count but we forgot to revise the title accordingly. We are sorry for your misunderstanding caused by this careless mistake.

Here, we also would like to explain why the third part was included in our first draft. Different feed additives, such as antibiotics, zinc oxide (ZnO), probiotics, and prebiotics, have been reported to induce shifts in the microbial community associated with growth performance, thereby providing new insights into helping identify functionally important microbes as prospective biomarkers that are beneficial for growth performance in pigs. The summary of changed microbiota will help to find the potential target microbiota for antibiotics or ZnO, and therefore, to help find the alternatives of antibiotics or ZnO in swine feed. Another reviewer also asked questions similar to yours. Here, we would like to show the third section presented in our first draft:

“Promoting host performance by modulating the gut microbiota: the role of feed-additives”

Feed additives are widely used in swine production to promote growth performance. Nutritive additives, such as therapeutic levels of Zinc oxide (ZnO), or non-nutritive additives, such as antibiotics, probiotics and prebiotics, promote the growth performance and/or the intestinal health of pigs by modulating the composition and function of gut microorganisms. Distinct bacterial diversity, composition, and metabolites can be observed in the gut of pigs offered diets with different feed additives (Table S5-S7).

In-feed antibiotics and gut microbiota in pigs

Over the past decades, in-feed antibiotics were recognized as classic growth promoters and were commonly used in the swine industry [51]. Although the modulation of in-feed antibiotics on the gut microbiota is type-dependent [8, 9, 11, 13, 15], studies show that the proportion of *Prevotella* [8, 11, 13], *Succinivibrio* [11], *Faecalibacterium* and *Roseburia* [13] can be increased by the majority of antibiotics supplemented into the diet (Table S5). As mentioned above, *Prevotella* is a dominant and well-studied genus in pig GIT. The strong capability of *Prevotella* to degrade ingredients, such as hemicellulose and pectin, is important for the maintenance of intestinal health and energy supply which accounts for up to 30% of gross energy in the colon of pigs. The increased abundance of *Prevotella* due to the treatment of in-feed antibiotics indicates a stimulated development and maturation of “adult-like” microbial community [9]. These bacteria are metabolically complementary to each other, as *Prevotella* produces acetate, while *Roseburia* and *Faecalibacterium* produce butyrate by consuming acetate [55, 86]. Age may be a key factor influencing the effect of in-feed antibiotics on the gut microbiota. Dietary supplementation with gentamicin at 5 mg/d only increased the abundance of *Prevotella* and *Lactobacillus* in the gut of piglets at day 28 and 42, but not at day 7 [15]. Tylosin at 40 mg/kg was found to enrich *Prevotella* in the gut of finisher pigs at days 70 and 133, but an opposite effect was observed in pigs at day 91 and day 154 [9]. Similar changes are also observed in the abundance of *Lactobacillus* [9].

The effect of in-feed antibiotics on the gut microbiota also varies depending on intestinal segment. After exposure to an antibiotic cocktail of 100 mg/kg chlortetracycline, 100 mg/kg sulfamethazine, and 50 mg/kg penicillin, *Streptococcus* in the ileum, cecum and colon of 90-day old growing pigs is decreased, but *Treponema* is decreased only in the cecum and colon [12]. In addition, *Helicobacter* spp. is decreased whereas *Anaeroplasma* and *Paraprevotella* spp. are increased in the cecum, and *Turicibacter* is decreased in the colon [12]. The impact of in-feed antibiotics on the gut microbiota of pigs also show differences between the foregut (stomach, duodenum, jejunum and ileum) and the hindgut (cecum and colon) [7, 14, 87]. Typically, the abundance of some bacteria, such as *Clostridium*, *Bacillus*, and *Sharpea*, are consistently decreased in the foregut, but not in the hindgut by an antibiotic cocktail of 50 mg/kg olaquinox, 50 mg/kg oxytetracycline calcium, and 50 mg/kg kitasamycin [7]. Moreover, the abundance of *Prevotella* in the hindgut, but not in the foregut, can be increased by chlortetracycline at 100 mg/kg [14].

Other feed additives and microbiota in pig GIT

The over-reliance on in-feed antibiotics raised serious problems, such as dysbiosis, drug residues, and resistant strains in the animal gut, which resulted in the ban on the routine use of antibiotics in livestock feed [6, 51]. In this case, other feed additives with similar positive effect of antibiotics attract much attention.

The anti-diarrhea role of therapeutic levels of zinc oxide (ZnO) has been proven in weaning piglets [16, 87, 88]. Current studies suggest that piglets offered diets with 2000~3000 mg/kg of ZnO harbor similar microbiota compared to those in-feed antibiotics-treated pigs (Table S6). However, intestinal location-specific alterations of gut microbiota are observed in piglets treated with therapeutic levels of ZnO [87, 88]. Notably decreased abundance of *Bacteroides* and *Clostridium*, and increased *Prevotella*, *Lactobacillus* and *Streptococcus* are also found in the feces of pigs received 2500 mg/kg of dietary ZnO at day 10 and day 21 after weaning [16].

Probiotics are living microorganisms, which contribute a health benefit to the host after administered in adequate amounts [89]. Probiotics and prebiotics are utilized to intervene or regulate the microecosystem in the GIT of pigs, especially during weaning (Table S7). Bacteria of *Lactobacillus* are the most commonly used probiotics in swine industry [14]. However, the influence of such probiotics on the whole microbial profile in pig GIT is rarely reported. Limited studies show that dietary supplement of *L. reuteri* increases the abundance of several genera including *Treponema*, *Parabacteroides*, and *Desulfovibrio* in the cecum and colon of 40-day old pigs [14]. The abundance of *Lactobacillus*, *Streptococcus*, *Prevotella*, and *Staphylococcus* in the jejunum of pigs is also increased by the treatment of *L. reuteri* compared to chlortetracycline (100 mg/kg) [14]. Prebiotics are selectively fermented feed ingredients, which confer a beneficial physiological effect on the host by making changes both in the composition and/or activity of gut microbiota [90]. Interestingly, the modulation of microbiota by prebiotics, such as some dietary fibers, may be similar to what is observed in carbadox, an in-feed antibiotic [91]. A strong association is shown between *Prevotella* and fiber-rich diets [39, 92]. An increase of soluble fiber or non-starch polysaccharides in the diet leads to the enrichment of *Prevotella* spp. and subsequently increase of butyryl-coenzyme A (CoA) transferase gene expression in the feces of

nursery pigs [91]. Chitosan, products of N-deacetylation of chitin, is used as a prebiotic in pigs because of its various biological activities [17]. The abundance of *Prevotella* in the cecum of 49-day old pigs can be increased by low-molecular-weight chitosan (LC) compared to control. Meanwhile, the abundance of *Succinivibrio*, a bacterial genus involved in the biodegradation of polysaccharides, is increased 38 folds [17], which contributes the declined pH in cecum due to the production of acetate and succinic acid [17, 93]. Accompanying the alteration of the gut microbiota, pathways related with carbohydrate metabolism (15%), metabolism of cofactors and vitamins (9%), energy metabolism (8%), and glycan biosynthesis and metabolism (6%) are also enriched by LC supplement [17], suggesting that the proliferation of these bacteria may help to degrade complex carbohydrates. Similar change in *Prevotella* in the ileum of pigs offered dietary chitooligosaccharides at 500 mg/kg is also observed in another study [94].

These studies characterize the shifts of microbial community associated with growth performance in pigs due to different in-feed intervention strategies. The enrichment of specific microbes, supported by beneficial functionality, can help to identify functionally “key” microbes to improve the performance and/or intestinal health of host.

We are sorry that we could not put this part into our manuscript due to the limited word count.

We hope these explanations are detailed enough. If you have any other questions, please do not hesitate to ask and we will try our best to answer them.

3. Several recent papers have provided comprehensive reviews related to gut microbiota and pigs. Most information in the present manuscript can be found in following papers. The present manuscript does not provide a new insight and is also lack of deep discussion regarding this topic.

Y Patil, R Gooneratne, XH Ju. Interactions between host and gut microbiota in domestic pigs: a review- Gut Microbes, 2020.

ND Aluthge, DM Van Sambeek et al. BOARD INVITED REVIEW: The pig microbiota and the potential for harnessing the power of the microbiome to improve growth and health- Journal of animal Science. 2019.

Raphaële Gresse,Frédérique Chaucheyras-Durand, et al. Gut Microbiota Dysbiosis in Postweaning Piglets: Understanding the Keys to Health. Trends in Microbiology. 2017.

Response: thanks a lot for your comments. We totally understand your query. We also agree that there is more or less similarity between our review and the others, but we think the biggest difference between our manuscript and the published articles is the focus (different key point). We do not only summarize the microbial composition in swine gut, but also try to screen some potential functional microbiota for further research.

Therefore, we first summarize the composition of fecal microbiota in pigs with different age, helping to draw a full picture of the dynamic distribution of gut microbiota in pigs, which could be a reference of the succession of microbial community in healthy pigs. In order to further verify this summary, we then conduct the meta-analysis under strict restrictions and some meaningful findings can be found: the weaning period may be a powerful window to regulate the gut microbiome in pigs; A “core” microflora composed of 19 genera can be found in more than 90% of samples. Accordingly, *Bacteroides*, *Escherichia* and *Lactobacillus* are predominant in lactating pigs, while *Prevotella*, *Lactobacillus* and *Oscillispira* are dominant in nursery pigs. In growing pigs, the three most abundant genera are *Prevotella*, *Lactobacillus* and *Faecalibacterium*, and the top 3 genera are *Prevotella*, *Lactobacillus* and *Streptococcus* in finishing pigs. These “core” bacterial groups may be the potential growth performance and gut health beneficial microbiota, or the potential target microbiota for nutritional or non-nutritional regulation. In the section “Contribution of the gut microbiota to host phenotypes”, it is amazing that some of the host phenotype contributors belongs to the defined “core” bacteria, which is worthy of further investigation. Although we have comprehensively summarized the dynamic shifts of gut microbiota in pigs, substantial work remains to investigate metabolites as the functional output of microorganisms to interact with the host phenotype. It is

also possible to achieve the desired phenotype through effective regulation of constantly changing microbial community or metabolites. In fact, this review was summarized and formed in the process of designing our animal experiments. At present, we have completed the research on intestinal symbiotic bacteria and the gut health of weaned piglets, and excavated the target bacterial metabolites that can significantly promote intestinal barrier and immune function of piglets (manuscript is in preparation). These studies were designed based on the inspiration of the current review, which also proves the correctness of our views presented in the current manuscript. Therefore, we believe that further research targeting these “core” microbes and the dynamic distribution of microbiota, as well as the related function is of great importance in swine production.

However, we appreciate your queries and references. We are also very sorry that these references were not cited in our original manuscript. Therefore, we have cited these articles in our revised manuscript and made a brief discussion to show the differences between our review and the others (Line 339-348). Thank you again for these helpful suggestions. Any other possible questions you may have are welcome. We thank you for all your work and efforts to improve our manuscript.

Reviewer #2 (Comments for the Author):

The review by Luo and Ren et al describes in detail a meta-analysis geared towards identifying patterns in the porcine gut microbiota. The manuscript is topical, well-written, the methods are adequate and the results are well detailed.

Response: thank you very much for your positive comments and support. The manuscript has been carefully revised in accordance with your comments. Details are listed below.

A few comments that would improve the manuscript:

- The main question here is well known by everybody working in the microbiome area. Are the microbiome changes a cause or an effect? The authors need to include statements that qualify this, if possible...

Response: thank you for your professional suggestion. The question whether the microbiome changes a cause or an effect is really interesting, but it's very difficult to qualify this as far as we know. On the one hand, recent studies on both humans and animals suggest that microorganisms can be found in the fetus, placenta, amniotic fluid or uterus (Mayer et al., 2012; Karstrup et al., 2017; Hemberg et al., 2015; Perez-Muñoz et al., 2017), which indicates that microbial colonization has already started before parturition. This hypothesis has been confirmed, for example, in the study conducted on cattle by Alipour et al. (2018), who also confirmed that the microbiota in the gastrointestinal tract is subject to rapid changes after birth. On the other hand, microbiota in the gut of pigs can be influenced by diet and environmental factors. In the first week after birth, *Bacteroides*, *Escherichia* and *Clostridium* are the three most abundant genera. The shift from sow milk to solid feed helps shape the microbiota in gastrointestinal tract of piglets. A marked shift from *Bacteroides* to *Prevotella* has been observed as piglets age/grow. Overall, we prefer to conclude that the changes in gut microbiome can be both a cause or an effect.

- Mayer, M., et al. Development and genetic influence of the rectal bacterial flora of newborn calves. *Vet. Microbiol.* 2012, 161, 179–185.
- Karstrup, C.C. et al. Presence of bacteria in the endometrium and placentomes of pregnant cows. *Theriogenology* 2017, 99, 41–47.
- Hemberg, E. et al. Occurrence of bacteria and polymorphonuclear leukocytes in fetal compartments at parturition; relationships with foal and mare health in the peripartum period. *Theriogenology* 2015, 84, 163–169.
- Perez-Muñoz, M.E. et al. A critical assessment of the “sterile womb” and “in utero colonization” hypotheses: Implications for research on the pioneer

infant microbiome. *Microbiome* 2017, 5, 48.

- Alipour, M.J. et al. The composition of the perinatal intestinal microbiota in cattle. *Sci. Rep.* 2018, 8, 10437.

• The authors also need to provide some practical advice, since there is a danger of well-prepared manuscripts such as this one to be highly theoretical. The one question I would like addressed here is what would be the main practical outcome of this manuscript? If this would have to be boiled down to one paragraph that would help a pig producer to get a better value (and better welfare) for his pigs, what would that paragraph be?

Response: thank you for your nice comments. We totally agree that one paragraph about practical outcome should be focused. The main practical outcome of this manuscript is that pig producer should not only feed pigs, but also feed the age associated gut microbiome, weaning period may be a powerful window to regulate the gut microbiome in pigs, and the potential target microbiota may probably be the “core” microbiome (such as *Bacteroides*, *Escherichia* and *Lactobacillus* et al.).

Actually, there are several places where this manuscript mentioned practical outcome:

Line 34-39: A number of 19 bacterial genera, including *Bacteroides*, *Prevotella* and *Lactobacillus* can be found in more than 90% of pigs and three enterotypes can be identified in all pigs at different ages, suggesting there is a “core” microbiota in the gut of healthy pigs, which can be a potential target for nutrition- or health-regulation. The “core” members benefit the growth and gut health of the host. These findings help to define an “optimal” gut microbial profile for assessing, or improving, the performance and health status of pigs at different growth stages.

Line 69-73: On the other hand, different feed additives, such as antibiotics, zinc oxide (ZnO), probiotics, and prebiotics, have been reported to induce shifts in

the microbial community associated with growth performance, thereby providing new insights into helping identify functionally important microbes as prospective biomarkers that are beneficial for growth performance of pigs.

Line 466-468: Such findings improve our understanding of how the gut microbiome influences porcine FE. Characterizing “FE-associated” microbial biomarkers may help to define an “optimal” microbial profile to improve the FE and growth performance of pigs.

Line 476-511: “Conclusion” section. We also revised this section to make it more practical. Please see the revised contents in the manuscript.

We believe the paragraph from line 302-310 may help a pig producer to get a better value (and better welfare). This paragraph shows the “core” microbiome which selected according to meta-analysis, these “core” microbiome may be the potential next generation probiotics or the potential target regulation microbiome in our opinion.

• My view is that resilience of the microbiome (and the animal as a whole) is the one aim of pig production. Can the authors comment on this from this meta-analyses? I agree that resilience can mostly be assessed through a side-by-side comparison and a change in some experimental factors...

Response: thank you for your comments. The resilience of the microbiome is an interesting topic and have practical meaning. However, our meta-analysis is unable to comment on this. There are too much factors which influence the microbial resilience, current studies haven’t set a scientific design on this topic. In order to conduct the current meta-analysis, we have screened 16 studies from the 62 references due to the consideration of cut down the influence of other factors, which makes the meta-analysis represent the dynamic of changes in microbiota, but not the resilience. We believe that a side-by-side comparison will be very useful to show/compare the resilience of the microbiome in swine gut.

- Are there details of medicated feed available in any of the studies reviewed here, since medicated and even zinc oxide feed would have an effect on the microbiome. And in light of the 2022 changes to the zinc oxide legislation in Europe, are these results applicable to the post zinc oxide ban?

Response: thank you for your nice comments. As we replied to another reviewer, our first draft of this manuscript contains another section, ways to promote host performance by modulating the gut microbiota using feed additives. This section was deleted due to the limited word count.

Below are the related contents that has been deleted:

“Promoting host performance by modulating the gut microbiota: the role of feed-additives”

Feed additives are widely used in swine production to promote growth performance. Nutritive additives, such as therapeutic levels of Zinc oxide (ZnO), or non-nutritive additives, such as antibiotics, probiotics and prebiotics, promote the growth performance and/or the intestinal health of pigs by modulating the composition and function of gut microorganisms. Distinct bacterial diversity, composition, and metabolites can be observed in the gut of pigs offered diets with different feed additives (Table S5-S7).

In-feed antibiotics and gut microbiota in pigs

Over the past decades, in-feed antibiotics were recognized as classic growth promoters and were commonly used in the swine industry [51]. Although the modulation of in-feed antibiotics on the gut microbiota is type-dependent [8, 9, 11, 13, 15], studies show that the proportion of *Prevotella* [8, 11, 13], *Succinivibrio* [11], *Faecalibacterium* and *Roseburia* [13] can be increased by the majority of antibiotics supplemented into the diet (Table S5). As mentioned above, *Prevotella* is a dominant and well-studied genus in pig GIT. The strong capability of *Prevotella* to degrade ingredients, such as hemicellulose and pectin, is important for the maintenance of intestinal health and energy supply which accounts for up to 30% of gross energy in the colon of pigs. The increased abundance of *Prevotella* due to the treatment of in-feed antibiotics indicates a stimulated development and maturation of “adult-like” microbial community [9]. These bacteria are metabolically complementary to each other, as *Prevotella* produces acetate, while *Roseburia* and

Faecalibacterium produce butyrate by consuming acetate [55, 86]. Age may be a key factor influencing the effect of in-feed antibiotics on the gut microbiota. Dietary supplementation with gentamicin at 5 mg/d only increased the abundance of *Prevotella* and *Lactobacillus* in the gut of piglets at day 28 and 42, but not at day 7 [15]. Tylosin at 40 mg/kg was found to enrich *Prevotella* in the gut of finisher pigs at days 70 and 133, but an opposite effect was observed in pigs at day 91 and day 154 [9]. Similar changes are also observed in the abundance of *Lactobacillus* [9].

The effect of in-feed antibiotics on the gut microbiota also varies depending on intestinal segment. After exposure to an antibiotic cocktail of 100 mg/kg chlortetracycline, 100 mg/kg sulfamethazine, and 50 mg/kg penicillin, *Streptococcus* in the ileum, cecum and colon of 90-day old growing pigs is decreased, but *Treponema* is decreased only in the cecum and colon [12]. In addition, *Helicobacter* spp. is decreased whereas *Anaeroplasma* and *Paraprevotella* spp. are increased in the cecum, and *Turicibacter* is decreased in the colon [12]. The impact of in-feed antibiotics on the gut microbiota of pigs also show differences between the foregut (stomach, duodenum, jejunum and ileum) and the hindgut (cecum and colon) [7, 14, 87]. Typically, the abundance of some bacteria, such as *Clostridium*, *Bacillus*, and *Sharpea*, are consistently decreased in the foregut, but not in the hindgut by an antibiotic cocktail of 50 mg/kg olaquinox, 50 mg/kg oxytetracycline calcium, and 50 mg/kg kitasamycin [7]. Moreover, the abundance of *Prevotella* in the hindgut, but not in the foregut, can be increased by chlortetracycline at 100 mg/kg [14].

Other feed additives and microbiota in pig GIT

The over-reliance on in-feed antibiotics raised serious problems, such as dysbiosis, drug residues, and resistant strains in the animal gut, which resulted in the ban on the routine use of antibiotics in livestock feed [6, 51]. In this case, other feed additives with similar positive effect of antibiotics attract much attention.

The anti-diarrhea role of therapeutic levels of zinc oxide (ZnO) has been proven in weaning piglets [16, 87, 88]. Current studies suggest that piglets offered diets with 2000~3000 mg/kg of ZnO harbor similar microbiota compared to those in-feed antibiotics-treated pigs (Table S6). However, intestinal location-specific alterations of gut microbiota are observed in piglets treated with therapeutic levels of ZnO [87, 88]. Notably decreased abundance of *Bacteroides* and

Clostridium, and increased *Prevotella*, *Lactobacillus* and *Streptococcus* are also found in the feces of pigs received 2500 mg/kg of dietary ZnO at day 10 and day 21 after weaning [16].

Probiotics are living microorganisms, which contribute a health benefit to the host after administered in adequate amounts [89]. Probiotics and prebiotics are utilized to intervene or regulate the microecosystem in the GIT of pigs, especially during weaning (Table S7). Bacteria of *Lactobacillus* are the most commonly used probiotics in swine industry [14]. However, the influence of such probiotics on the whole microbial profile in pig GIT is rarely reported. Limited studies show that dietary supplement of *L. reuteri* increases the abundance of several genera including *Treponema*, *Parabacteroides*, and *Desulfovibrio* in the cecum and colon of 40-day old pigs [14]. The abundance of *Lactobacillus*, *Streptococcus*, *Prevotella*, and *Staphylococcus* in the jejunum of pigs is also increased by the treatment of *L. reuteri* compared to chlortetracycline (100 mg/kg) [14]. Prebiotics are selectively fermented feed ingredients, which confer a beneficial physiological effect on the host by making changes both in the composition and/or activity of gut microbiota [90]. Interestingly, the modulation of microbiota by prebiotics, such as some dietary fibers, may be similar to what is observed in carbadox, an in-feed antibiotic [91]. A strong association is shown between *Prevotella* and fiber-rich diets [39, 92]. An increase of soluble fiber or non-starch polysaccharides in the diet leads to the enrichment of *Prevotella* spp. and subsequently increase of butyryl-coenzyme A (CoA) transferase gene expression in the feces of nursery pigs [91]. Chitosan, products of N-deacetylation of chitin, is used as a prebiotic in pigs because of its various biological activities [17]. The abundance of *Prevotella* in the cecum of 49-day old pigs can be increased by low-molecular-weight chitosan (LC) compared to control. Meanwhile, the abundance of *Succinivibrio*, a bacterial genus involved in the biodegradation of polysaccharides, is increased 38 folds [17], which contributes the declined pH in cecum due to the production of acetate and succinic acid [17, 93]. Accompanying the alteration of the gut microbiota, pathways related with carbohydrate metabolism (15%), metabolism of cofactors and vitamins (9%), energy metabolism (8%), and glycan biosynthesis and metabolism (6%) are also enriched by LC supplement [17], suggesting that the proliferation of these bacteria may help to degrade complex carbohydrates. Similar change in *Prevotella* in the ileum of pigs offered dietary chitooligosaccharides at 500 mg/kg is also observed in another study [94].

These studies characterize the shifts of microbial community associated with growth performance in pigs due to different in-feed intervention strategies. The enrichment of specific microbes, supported by beneficial functionality, can help to identify functionally “key” microbes to improve the performance and/or intestinal health of host.

If you agree or like, we can also put this section back to our manuscript. However, it is very difficult to include this section due to the limited word count.

Specific comments

L69 - I would like to see the materials and methods here, not in the supporting material

Response: thank you for the suggestion. For the method of meta-analysis, the description of methods often warrants standardized formulations, which will cause high similarity to previously published work. We did follow your suggestions and added the materials and methods in the manuscript, but our manuscript was returned to us at 2021.12.01 after we submitted the revised manuscript. The editors told us “*Microbiology Spectrum* screens all submitted manuscripts using Cross-check (A program that looks for text similarities with other digital sources) and it has been brought to my attention that portions of the text in your manuscript Spectrum00688-21 may occur in publicly available publications and documents (please see the attachment)”, which is specific to the method of meta-analysis.

We think the best way is to refer to this previous study on the meta-analysis method, so in the manuscript we revised the sentences as “The meta-analysis was carried out following the previously described method (67). A total of 16 studies from previous 63 papers were included in the meta-analysis and are described in Table S1.”

67. Holman DB, Brunelle BW, Trachsel J, Allen HK, Bik H. 2017. Meta-analysis to define a core microbiota in the swine gut. *mSystems* 2:e00004-17.

L69 - please include an age threshold for each production stage (pre-weaning-nursery-growth-finish)

Response: thank you for your nice comments. We have included all the age threshold in the manuscript. You can also find the details below:

Line 103 Development of the gut microbiota in piglets from birth to weaning (0-20 d);

Line 152 Development of gut microbiota in piglets from weaning to 7 d post weaning (21-28 d);

Line 211 Development of gut microbiota in pigs from days 28 to 154 (28-154 d)

L76 - please clarify is the nursery is considered the post-weaning period

Response: thank you for your comments. According to the reference, the nursery here isn't including the post-weaning period, we then add "(0-21 d)" here to help the reader better understanding. You can find the details in Line 89.

L88 - can the stability of the microbiome be calculated?

Response: thank you for your comments. The stability of the microbiome can't be calculated, it's just a description. To avoid misunderstanding and consider the exactness of word in our manuscript, we have deleted this.

L137 - insert anti-inflammatory instead of ant-inflammatory

Response: thanks a lot for your suggestion. We have revised this word to "anti-inflammatory". In case there are similar problem, we also checked similar word in the whole manuscript.

L153 - this may be of use when discussing Prevotella Intestinal microbiota of pigs determines the response to vaccines - Articles - pig333, pig to pork community

Response: thank you for your comments. It's a pity that we can't open the article: <https://www.pig333.com/search/3/0/index.php?keyword=Intestinal%20microbiota%20of%20pigs>

L174 - does adult like mean? stable?

Response: thanks a lot for your comments. Yes, the adult like means a relative stable microbiota composition like the adult pigs. We use this word referencing to other papers.

L200 - how do you objectively assess stability?

Response: thanks a lot for your comments. Seriously, it's not rigorous to objectively assess stability and the stability of the microbiome can't be calculated. Our word here is just a description, so we delete this.

L225 - not sure about the intricate details of the meta-analysis, but sometimes, an age-related analysis can be distorted by extreme age values. For example if you only have a few pigs that are 170-day old pigs (compared to many more at weaning-nursery), any specific bacteria for those pigs are at a very low level, but not due to the fact that the bacteria are indeed low at that age, but because there is only a few pigs represented. Just wanted to alert the authors to this aspect...

Response: it's a nice reminding and we really appreciate this. It's true that more studies were focused on the younger pigs (lactation and nursery period). In the current meta-analysis, there are 367 samples for growing period (61-116 d) and 126 samples for finishing period, both sample size are larger than 100, which make the meta-analysis of these period meaningful and representative.

L249 - replace top3 with top 3

Response: thanks a lot for your suggestion. We have revised the word "top3" to "top 3" in the manuscript (Line 307).

L267 - please clarify the hypothesis - it would make sense that Escherichia are contributors to early life diarrhoeas, what is the novelty outlined here?

Response: thanks a lot for your comments. According to the meta-analysis here,

there is high abundance of *Escherichia* in the gut of healthy sucking piglets and one of the commensal microbiota. It's not the general knowledge that dietary or environmental challenge cause the high abundance of pathogens such as *Escherichia*, and also not the general knowledge that diarrhea piglets contain high abundance of *Escherichia*. We therefore guess the high abundance of *Escherichia* is a cause factors other than effect results of diarrhea.

L319 - cause and effect issue here, if the authors have any comments on discerning whether the microbiome changes are a cause of an effect, they should be highlighted there

Response: thanks a lot for your nice comments. We will treat this a cross-talk effect other than a cause of an effect. The interaction between microbiota and host is very complex, it can be both a cause and effect issue. Our review here is try to summary the relationship between some target microbiome and the host phenotypes, this will be the foundation of further research: whether it's possible to achieve target growth performance by regulating these microbiota (a cause effect), or as a biomarker to indicate the target growth performance or health status (an result of effect). We also added this in line 469-474.

L362 - for this paragraph, the authors need to explain the relationship between feed efficiency (higher is better), FCR (lower is better) and RFI (lower is better), to clarify the data that are explained in the paragraph

Response: thanks a lot for your suggestions. We have added these clarification in the manuscript.

L399 - as outlined above, can the authors include some practical recommendation for pig producers? What would make up a good/stable/resilient microbiome? Feed, add/remove medication, additives, appropriate care, a mix of these factors? In my view, if producers really care for their animals, they will make sure, they are looked

after, they will offer them good quality feed, good quality accommodation, good healthcare and then the microbiome will follow...any opinions?

Response: thanks a lot for your comments. To make up a good/stable/resilient microbiome for pigs, a system consideration will be needed. Besides offer them good quality feed, good quality accommodation, good healthcare et al., we can also feed the target microbiome. Because the cecum/colon microbiota use the undigested nutrients to growth and keep their abundance, we suggest to add some substrate of the target microbiota preferring in the pig feed, this will help to increase the abundance of the specific microbiota and benefit the gut health of pigs. We also add this in our manuscript.

Staff Comments:

Preparing Revision Guidelines

- Point-by-point responses to the issues raised by the reviewers in a file named "Response to Reviewers," NOT IN YOUR COVER LETTER.

Response: thanks a lot for your comments. We have finished the responses to the reviewers and you can find the details above.

- Upload a compare copy of the manuscript (without figures) as a "Marked-Up Manuscript" file.

Response: thanks a lot for your comments. The “Marked-Up Manuscript” has

been uploaded separately.

- Each figure must be uploaded as a separate file, and any multipanel figures must be assembled into one file.

Response: thanks a lot for your comments. We have followed the suggestion.

- Manuscript: A .DOC version of the revised manuscript

Response: thanks a lot for your comments. You can find the revised manuscript with .DOC version.

- Figures: Editable, high-resolution, individual figure files are required at revision, TIFF or EPS files are preferred

Response: thanks a lot for your comments. We have uploaded all the individual figure files. All our figures are formatted in TIFF.

January 19, 2022

Prof. Daiwen Chen
Institute of Animal Nutrition, Sichuan Agricultural University, Chengdu, Sichuan
chengdu
China

Re: Spectrum00688-21R1 (Dynamic distribution of gut microbiota in pigs at different growth stages: composition and contribution)

Dear Prof. Daiwen Chen:

Link Not Available

Sincerely,

Cheng-Yuan Kao

Journals Department
Reviewer comments:

Reviewer #1 (Comments for the Author):

Thank you for your careful response. A recent paper "Meta-analysis To Define a Core Microbiota in the Swine Gut" published in mSystems performed a meta-analysis using 20 publically available data sets from high-throughput 16S rRNA gene sequence studies of the swine gut microbiota, which is very similar to your manuscript. They also defined the core microbiota. Additionally, the paper conducted the weighted and unweighted UniFrac distances and Bray-Curtis and Binary-Jaccard dissimilarities analysis as well. They also found that age of pigs significantly influences the gut microbiota. This paper should be included in your discussion.

Reviewer #3 (Comments for the Author):

Reviewer comments

Manuscript: Spectrum00688-21R1 Dynamic distribution of gut microbiota in pigs at different growth stages: composition and

contribution

The authors analyzed the bacterial community in the gut of pigs at different growth phases based on the data reported in 63 published papers. The rRNA gene sequences from untreated, healthy pigs from 16 different studies from GenBank is used for a meta-analysis to verify characteristic microbial populations. Based upon the metadata, the contribution of the gut microbiota to growth phenotypes of the host were evaluated, in order to explain functional microbes, and potential biomarkers in swine.

The data analysis methods are correct.

The English of the text is well written and well readable but needs additional checking with a professional translator.

The uniqueness of the text is more than 90% by AntiPlagiarism.NET.

The text contains some misspellings and typos.

There are some comments:

Line 30: age >> ages

Line 42 - antibiotic >> antibiotics

Line 43: urging researchers and pig producers search new >> urge researchers and pig producers to search for new

Line 58: is influenced >> are influenced

Line 58: After sentence "The diversity, composition and function of gut microbial community is influenced by various factors including diet, age, stress, and the environment." add citation (Danilenko et al., 2021). Please add to the reference - Danilenko, V.N.; Devyatkin, A.V.; Marsova, M.V.; Shibilova, M.U.; Ilyasov, R.A.; Shmyrev, V.I. Common inflammatory mechanisms in COVID-19 and Parkinson's diseases: the role of microbiome, pharmabiotics and postbiotics in their prevention. Journal of Inflammation Research 2021, 14, 6349-6381. doi: 10.2147/JIR.S333887

Line 61: have been >> has been

Line 108: similarly >> similar

Line 142: Escherichia. coli >> Escherichia coli

Line 168: increased >> is increased

Line 181: are dominated >> is dominated

Line 184: an "mature-like microbiota" >> a "mature-like microbiota"

Line 227: on the characteristics >> of the characteristics

Line 248: lifetime >> life

Line 260: maybe >> may be

Line 275: shows over-growth >> show over-growth

Line 300: other three >> the other three

Line 304: in healthy pig >> in healthy pigs

Line 311: shown >> that show

Line 311: less similarity >> fewer similarities

Line 317: swine gut of different age >> swine gut of different ages

Line 324: was discussed >> were discussed

Line 325: digesta >> digestion

Line 325: were found >> was found

Line 329: difference of microbial >> difference in microbial

Line 331: may be as a key >> may be a key

Line 350: piglets from day >> piglets from days

Line 364: feed intake >> food intake

Line 368: feed intake >> food intake

Line 371: feed intake >> food intake

Line 372: feed intake >> food intake

Line 373: still remains >> remains

Line 376: affiliated to >> affiliated with

Line 414: orthologies >> orthologs

Line 423: these microbiota >> this microbiota

Line 436: antibiotic >> antibiotics

Line 438: urging researchers and pig producers search new >> urge researchers and pig producers to search for new

Line 438: alternatives >> alternative

Line 457: community >> communities

In this paper "Tsuchida, S.; Maruyama, F.; Ogura, Y.; Toyoda, A.; Hayashi, T.; Okuma, M.; Ushida, K. Genomic Characteristics of Bifidobacterium thermacidophilum Pig Isolates and Wild Boar Isolates Reveal the Unique Presence of a Putative Mobile Genetic Element with tetW for Pig Farm Isolates. Frontiers in microbiology 2017, 8, 1540-1540. doi: 10.3389/fmicb.2017.01540" Authors described that pigs do carry Bifidobacterium spp. as the primary component of intestinal microbiota, at least seven strains of Bifidobacterium thermacidophilum. Why is there no information about pig Bifidobacterium in your manuscript?

Please improve the manuscript according to the above comments.

No other comments.

Staff Comments:

Preparing Revision Guidelines

Please return the manuscript within 60 days; if you cannot complete the modification within this time period, please contact me. If you do not wish to modify the manuscript and prefer to submit it to another journal, please notify me of your decision immediately so that the manuscript may be formally withdrawn from consideration by Microbiology Spectrum.

Dynamic distribution of gut microbiota in pigs at different growth stages:
composition and contribution

Yuheng Luo^{1,†}, Wen Ren^{1,2,†}, Hauke Smidt³, André-Denis G. Wright⁴, Bing Yu¹, Ghislain
Schyns⁵, Ursula M. McCormack⁶, Aaron J. Cowieson⁵, Jie Yu¹, Jun He¹, Hui Yan¹, Jinlong Wu²,
Roderick I. Mackie⁷, and Daiwen Chen^{1,*}

¹ Key Laboratory for Animal Disease-Resistance Nutrition of Ministry of Education of China, Key
Laboratory for Animal Disease-Resistance Nutrition and Feed of Ministry of Agriculture of China, Key
laboratory of Animal Disease-resistant Nutrition of Sichuan Province, and Animal Nutrition Institute,
Sichuan Agricultural University, Chengdu, China, Chengdu 611130, People's Republic of China.

² DSM (China) Animal Nutrition Research Center Co., Ltd, Bazhou 065799, People's Republic of
China.

³ Laboratory of Microbiology, Wageningen University, Wageningen, Netherlands

[revised manuscript text omitted]

markedly decreased in the subsequent growth stages (Figure S7B and S7C, Table S6). Bacteria
belonging to the family Enterobacteriaceae, such as *E. coli*, often shows over-growth in the case of
dysbiosis (76) and may become the dominant bacteria, leading to the exacerbation of gut damage
and diarrhea (45, 77). Previous studies have focused on the enrichment of Enterobacteriaceae in
swine gut caused by dietary and environmental challenges, but our analysis provides another
hypothesis: the distinct high abundance of *Escherichia* in the GIT of suckling pigs may contribute
to the susceptibility of dysbacteriosis associated diarrhea. The high abundance of *Bacteroides*
during pre-weaning may be due to their ability to utilize monosaccharides and oligosaccharides
present in sows milk (35). Notably, *B. fragilis* is a symbiont found in the colon and colonizes
mucus or epithelium (78). It can protect the host from multiple preclinical colitis via the induction
of the anti-inflammatory response, such as increased interleukin-10 production by Foxp³⁺
regulatory T cells (79, 80). In addition, a dramatic increase of *Prevotella* was found post-weaning,
and was the most abundant genera in the gut of nursery (22.32%), growing (27.79%) and finishing
(15.25%) pigs (Figure S3B-D; Figure S4B; Table S6), suggesting that *Prevotella*, a class of
bacteria with powerful capacity to metabolize complex carbohydrates like hemicelluloses and
xylans (39, 40, 50), may be the dominant symbiont in the GIT of healthy post-weaning pigs.

The term “enterotype” is now recognized as an important characteristic of the gut microbiome. A
total of three enterotypes, E1 (*Bacteroides-Escherichia-Lactobacillus*), E2
(*Prevotella-Lactobacillus-Megasphaera*) and E3 (*Prevotella-Lactobacillus-Treponema*), were
identified from the collected sequences (Figure S8-S9). Among them, E1 accounted for
approximately 85% of the bacterial genera in the gut of suckling piglets (Figure S9A), while E3
accounted for 88% in finishing pigs (Figure S9D). For nursery and growing pigs, the division of

enterotypes was not absolute. The proportion of E1, E2 and E3 in the gut of nursery piglets was
16%, 50%, and 34%, respectively (Figure S9B). However, only two enterotypes, E2 (60%) and E3
(40%), were found in the gut of growing pigs (Figure S9C). *Bacteroides*, *Escherichia* and
*Lactobacillus* were the three main genera of bacteria contributing to E1 (Figure S10A, S10B and
S10D). *Megasphaera*, *Prevotella* and *Lactobacillus* were other three main genera contributing to

[revised manuscript text omitted]

Stages

- Finishing
- Growing
- Lactation
- Nursery

Enterotype Distribution

E1	85%
E2	8%
E3	7%

E1	16%
E2	50%
E3	34%

E1	0%
E2	60%
E3	40%

E1	1%
E2	11%
E3	88%

Contribute to Host Phenotypes

Body Weight

Food Intake

Feed Efficiency

E1 Bacteroides-Escherichia-Lactobacillus
 E2 Prevotella-Lactobacillus-Megasphaera
 E3 Prevotella-Lactobacillus-Treponema

Dear editors and referees,

Thanks a lot for giving us an opportunity to revise our manuscript again. We appreciate your patience and all your comments and suggestions that are very valuable and helpful to the improvement of our manuscript. Based on your comments, we re-checked our manuscript thoroughly and tried our best to settle your questions one by one. Here, we list the point-to-point response and enclose the revised manuscript as “Marked-up Manuscript” for your approval. Thank you again for all your efforts to improve our manuscript.

Kind regards,

Dr. Yuheng Luo on behalf of all authors

Point-to point responses

Reviewer #1 (Comments for the Author):

Thank you for your careful response. A recent paper "Meta-analysis To Define a Core Microbiota in the Swine Gut" published in mSystems performed a meta-analysis using 20 publically available data sets from high-throughput 16S rRNA gene sequence studies of the swine gut microbiota, which is very similar to your manuscript. They also defined the core microbiota. Additionally, the paper conducted the weighted and unweighted UniFrac distances and Bray-Curtis and Binary-Jaccard dissimilarities analysis as well. They also found that age of pigs significantly influences the gut microbiota. This paper should be included in your discussion.

Response: thanks a lot for your comments. We totally understand your query and agree that the content of the published article is more or less similar with our manuscript. After carefully considering your suggestion, we think the biggest difference between our manuscript and the published article is the focus (key point). We have also carefully read this published article and found that it mainly focuses on the microbial load along with pig GIT, while the focus of our discussion in this manuscript is the dynamic distribution of gut microbiota in

pigs at different growth stages, although the microbial load is something we could not avoid mentioning. That's why we cited this article as reference 68 in our manuscript: “In a recently published meta-analysis, the microbial load in the GIT and the dynamic shifts of the microbiota along with the GIT of pigs were discussed (68). A clear demarcation between the microbiota in the digestion samples from upper and lower GIT was found, with significantly higher diversity, richness, and evenness in the samples from the lower GIT (68). This finding is further supported by several later studies from 2017 to 2020, which confirm strong structural and functional differences in the colonized microbial populations between the upper and lower GIT (8, 11, 76, 77).” (Line 323-330). In addition, we should also note that the literatures cited in this article are published before 2017, and since then, several articles on the intestinal microbiome of pigs had been published. We appreciate your preciseness and professionalism, and thank you again for your suggestion.

Reviewer #2 (Comments for the Author):

Reviewer #3 (Comments for the Author):

Reviewer comments

Manuscript: Spectrum00688-21R1 Dynamic distribution of gut microbiota in pigs at different growth stages: composition and contribution

The authors analyzed the bacterial community in the gut of pigs at different growth phases based on the data reported in 63 published papers. The rRNA gene sequences from untreated, healthy pigs from 16 different studies from GenBank is used for a meta-analysis to verify characteristic microbial populations. Based upon the metadata, the contribution of the gut microbiota to growth phenotypes of the host were evaluated, in order to explain functional microbes, and potential biomarkers in swine.

Response: thank you very much for your comments and support. The manuscript has been carefully revised in accordance with your comments. Details are listed below.

The data analysis methods are correct.

The English of the text is well written and well readable but needs additional checking with a professional translator.

The uniqueness of the text is more than 90% by AntiPlagiarism.NET.

The text contains some misspellings and typos.

Response: thank you. There are native English speakers among the authors, and we have further revised the manuscript according to your suggestions. Please see the details listed below and the enclosed Marked-up Manuscript.

There are some comments:

Line 30: age >> ages

Response: it has been corrected as your suggestion (Line 30). Thank you.

Line 42 - antibiotic >> antibiotics

Response: it has been revised (Line 42). Thank you.

Line 43: urging researchers and pig producers search new >> urge researchers and pig producers to search for new

Response: the sentence has been improved as your comment (Line 43-44). Thank you.

Line 58: is influenced >> are influenced

Response: it has been corrected (Line 58). Thank you.

Line 58: After sentence "The diversity, composition and function of gut microbial community is

influenced by various factors including diet, age, stress, and the environment." add citation (Danilenko et al., 2021). Please add to the reference - Danilenko, V.N.; Devyatkin, A.V.; Marsova, M.V.; Shibilova, M.U.; Ilyasov, R.A.; Shmyrev, V.I. Common inflammatory mechanisms in COVID-19 and Parkinson's diseases: the role of microbiome, pharmabiotics and postbiotics in their prevention. Journal of Inflammation Research 2021, 14, 6349-6381. doi: 10.2147/JIR.S333887

Response: thanks a lot for your suggestion. This reference has been added as “reference 1” in the revised manuscript (Line 58).

Line 61: have been >> has been

Response: sorry for this error in grammar. It has been corrected as your kind suggestion (Line 61).

Line 108: similarly >> similar

Response: it has been revised (Line 108). Thank you.

Line 142: Escherichia. coli >> Escherichia coli

Response: it was a clerical error and has been corrected as your comment (Line 142). Thank you very much!

Line 168: increased >> is increased

Response: it has been improved (Line 168). Thank you.

Line 181: are dominated >> is dominated

Response: it has been corrected (Line 181). Thank you.

Line 184: an "mature-like microbiota" >> a "mature-like microbiota"

Response: it has been corrected (Line 185). Thank you.

Line 227: on the characteristics >> of the characteristics

Response: sorry for the misuse of prepositions. It has been corrected as your comment (Line 228). Thank you.

Line 248: lifetime >> life

Response: the word has been improved (Line 249) as your suggestion. Thank you.

Line 260: maybe >> may be

Response: it has been corrected (Line 261). Thank you.

Line 275: shows over-growth >> show over-growth

Response: it has been corrected (Line 276). Thank you.

Line 300: other three >> the other three

Response: thanks a lot for your comments. It has been corrected (Line 301).

Line 304: in healthy pig >> in healthy pigs

Response: thanks a lot for your comments. It has been corrected (Line 305).

Line 311: shown >> that show

Response: it has been improved (Line 312). Thank you.

Line 311: less similarity >> fewer similarities

Response: It has been corrected (Line 312). Thank you.

Line 317: swine gut of different age >> swine gut of different ages

Response: thanks a lot for your comments. It has been improved (Line 318).

Line 324: was discussed >> were discussed

Response: it has been improved as your kind suggestion (Line 325). Thank you.

Line 325: digesta >> digestion

Response: it has been revised (Line 326). Thank you.

Line 325: were found >> was found

Response: it has been corrected (Line 326). Thank you.

Line 329: difference of microbial >> difference in microbial

Response: the misused preposition has been modified according to your suggestion (Line 330). Thank you.

Line 331: may be as a key >> may be a key

Response: thanks a lot for your comments. It has been improved (Line 332).

Line 350: piglets from day >> piglets from days

Response: it has been corrected as your comment (Line 351). Thank you.

Line 364: feed intake >> food intake

Response: thanks a lot for your comment. We thought it might be better to use “feed intake” here than “food intake”. Because the object we discuss in this manuscript are pigs, not humans. Based on your comment, we referred to other articles and found that “food intake” can also be used for animals. Therefore, according to your suggestion, we changed the “feed intake” into “food intake” (Line 365).

Line 368: feed intake >> food intake

Response: it has been corrected (Line 368-369). Thank you.

Line 371: feed intake >> food intake

Response: it has been corrected (Line 372). Thank you.

Line 372: feed intake >> food intake

Response: it has been corrected (Line 373). Thank you.

Line 373: still remains >> remains

Response: it has been corrected (Line 374). Thank you.

Line 376: affiliated to >> affiliated with

Response: the preposition has been corrected as you kind suggestion (Line 377-378). Thank you.

Line 414: orthologies >> orthologs

Response: it has been corrected (Line 415). Thanks a lot!

Line 423: these microbiota >> this microbiota

Response: it has been revised (Line 424). Thank you.

Line 436: antibiotic >> antibiotics

Response: it has been corrected (Line 438). Thank you.

Line 438: urging researchers and pig producers search new >> urge researchers and pig producers to search for new

Response: the sentence has been improved as your comment (Line 439). Thank you.

Line 438: alternatives >> alternative

Response: it has been corrected (Line 440). Thank you.

Line 457: community >> communities

Response: it has been revised (Line 459). Thank you.

In this paper "Tsuchida, S.; Maruyama, F.; Ogura, Y.; Toyoda, A.; Hayashi, T.; Okuma, M.; Ushida, K. Genomic Characteristics of *Bifidobacterium thermacidophilum* Pig Isolates and Wild Boar Isolates Reveal the Unique Presence of a Putative Mobile Genetic Element with *tetW* for Pig Farm Isolates. *Frontiers in microbiology* 2017, 8, 1540-1540. doi: 10.3389/fmicb.2017.01540" Authors described that pigs do carry *Bifidobacterium* spp. as the primary component of intestinal microbiota, at least seven strains of *Bifidobacterium thermacidophilum*. Why is there no information about pig *Bifidobacterium* in your manuscript?

Response: thanks a lot for your professional question. Indeed, *Bifidobacterium* is one of the commensal bacterial genus in pig GIT, and we also include this information in our "Table S5. The relative abundance of microbial genera in the gut of pigs at each growth stage based on the collected sequences" (below is the screenshot of this table). However, we can learn from the results that the abundance of *Bifidobacterium* is not the top 10 genera, thus we didn't discuss *Bifidobacterium* in our manuscript.

Table S5. The relative abundance of microbial genera in the gut of pigs at each growth stage based on the collected sequences

Genus	Lactation, %	Nursery, %	Growing, %	Finishing, %
Escherichia	12.4481	1.1865	0.0555	0.5888
Lactobacillus	8.6713	6.2465	3.0027	4.4041
Prevotellaceae_Prevotella	5.7273	22.3193	27.7902	15.2487
Clostridiaceae_Clostridium	2.6426	0.6363	0.1051	3.0647
Bacteroides	15.5925	2.2629	0.0640	0.5838
Oscillospira	2.5190	2.8341	1.2773	1.7942
Fusobacterium	4.2181	0.7756	0.0006	0.5399
Paraprevotellaceae_Prevotella	0.7694	2.0456	2.0939	3.2970
Treponema	0.3506	1.1544	2.9578	2.0853
Bacillaceae_Bacillus	0.4710	0.8919	0.0044	0.0017
Streptococcus	1.2605	1.2482	4.2759	5.8591
Veillonella	0.6344	0.0566	0.0014	0.0075
Megasphaera	0.6619	2.3505	5.3453	2.6752
Phascolarctobacterium	1.1993	1.7962	1.3155	1.8754
Flexispira	0.1743	0.2863	0.0165	0.0107
Lachnospiraceae_Ruminococcus	2.7682	0.7492	0.2268	0.1009

Genus	Lactation, %	Nursery, %	Growing, %	Finishing, %
Lachnospiraceae_Clostridium	1.1702	0.1563	0.0484	0.0184
YRC22	0.0114	0.1744	0.3066	2.3919
Faecalibacterium	0.1524	2.1900	2.2010	0.5909
Dorea	0.7373	0.7271	0.4923	0.2939
Mitsuokella	0.1300	0.3504	0.3640	0.2005
Pasteurella	0.1600	0.0026	0.0000	0.0000
Trueperella	0.1953	0.0110	0.0006	0.0004
Mucispirillum	0.0013	0.0549	0.0103	0.0022
Bifidobacterium	0.4380	0.3469	0.2514	0.2317
Peptostreptococcus	0.2173	0.0073	0.0004	0.0324
Turicibacter	0.1416	0.0817	0.2430	1.1301
Erysipelotrichaceae_Clostridium	0.1445	0.0344	0.0001	0.0008
Butyrivimonas	0.5904	0.0791	0.0019	0.0019
Moryella	0.0056	0.0100	0.0000	0.0000

Please improve the manuscript according to the above comments.

No other comments.

Response: thank you again for all your comments and support.

Staff Comments:

Preparing Revision Guidelines

- Point-by-point responses to the issues raised by the reviewers in a file named "Response to Reviewers," NOT IN YOUR COVER LETTER.

Response: We have listed the point-by-point responses to the reviewers as your request. Please see the details above.

- Upload a compare copy of the manuscript (without figures) as a "Marked-Up Manuscript" file.

Response: The "Marked-Up Manuscript" has been enclosed separately as your request. Thank you very much for your processing.

- Each figure must be uploaded as a separate file, and any multipanel figures must be assembled into one file.

Response: The figures have been prepared and uploaded as your request. Thank you.

- Manuscript: A .DOC version of the revised manuscript

Response: The revised manuscript with .DOC version has been included. Thank you.

- Figures: Editable, high-resolution, individual figure files are required at revision, TIFF or EPS files are preferred

Response: We guarantee that the format of all pictures meets the requirements of Microbiology Spectrum. Thank you.

March 6, 2022

Prof. Daiwen Chen
Institute of Animal Nutrition, Sichuan Agricultural University, Chengdu, Sichuan
chengdu
China

Re: Spectrum00688-21R2 (Dynamic distribution of gut microbiota in pigs at different growth stages: composition and contribution)

Dear Prof. Daiwen Chen:

Your manuscript has been accepted, and I am forwarding it to the ASM Journals Department for publication. You will be notified when your proofs are ready to be viewed.

Sincerely,

Cheng-Yuan Kao
Editor, Microbiology Spectrum

Journals Department
Supplemental Figures and Tables: Accept